# PoE-World: Compositional World Modeling with Products of Programmatic Experts

**Wasu Top Piriyakulkij**[1]    **Yichao Liang**[2]    **Hao Tang**[1]
**Adrian Weller**[2,3]    **Marta Kryven**[4]    **Kevin Ellis**[1]
Cornell University[1]    University of Cambridge[2]
The Alan Turing Institute[3]    Dalhousie University[4]

## Abstract

Learning how the world works is central to building AI agents that can adapt to complex environments. Traditional world models based on deep learning demand vast amounts of training data, and do not flexibly update their knowledge from sparse observations. Recent advances in program synthesis using Large Language Models (LLMs) give an alternate approach which learns world models represented as source code, supporting strong generalization from little data. To date, application of program-structured world models remains limited to natural language and grid-world domains. We introduce a novel program synthesis method for effectively modeling complex, non-gridworld domains by representing a world model as an exponentially-weighted product of programmatic experts (PoE-World) synthesized by LLMs. We show that this approach can learn complex, stochastic world models from just a few observations. We evaluate the learned world models by embedding them in a model-based planning agent, demonstrating efficient performance and generalization to unseen levels on Atari's Pong and Montezuma's Revenge. We release our code and display the learned world models and videos of the agent's gameplay at `https://topwasu.github.io/poe-world`.

## 1 Introduction

How should an intelligent agent represent the dynamics of the natural world? We want a representation that is efficiently learnable, yet flexible enough to handle stochasticity and partial observability, and which supports planning and decision-making. Neural network world models such as Dreamer [1] are radically flexible, but demand enormous training data (compared to humans [2]). Symbolic world models such as WorldCoder [3] instead generate a Python program to represent how the world works. These programmatic world models are data-efficient, because program synthesis requires less data than neural network training— but struggle to scale beyond simple gridworlds, as they do a discrete combinatorial search to find a single large program describing everything about how the world works.

We take inspiration from a longstanding view in philosophy and cognitive science of the mind as a community of interacting experts [4, 5, 6, 7]. This modular organization is evident across multiple scales in natural intelligence, from functional specialization of brain systems [8] to the distinct learning trajectories of specific skills [9, 10]. We integrate this modular perspective with the computational paradigm of learning as Program Synthesis, which models learned concepts as symbolic programs [11, 12, 13, 14, 15, 16]. We extend this line of work by proposing a new computational account of learning world models: as the acquisition of context-specific expert programs, which are refined through practice and reused compositionally to support flexible, goal-directed behavior.

Algorithmically, our key idea is to *decompose the problem of learning a world program into learning hundreds of small programs*. Each of these learned programs encodes a different causal law, which we probabilistically aggregate to predict future observations (Figure 1a). This makes our world

39th Conference on Neural Information Processing Systems (NeurIPS 2025).

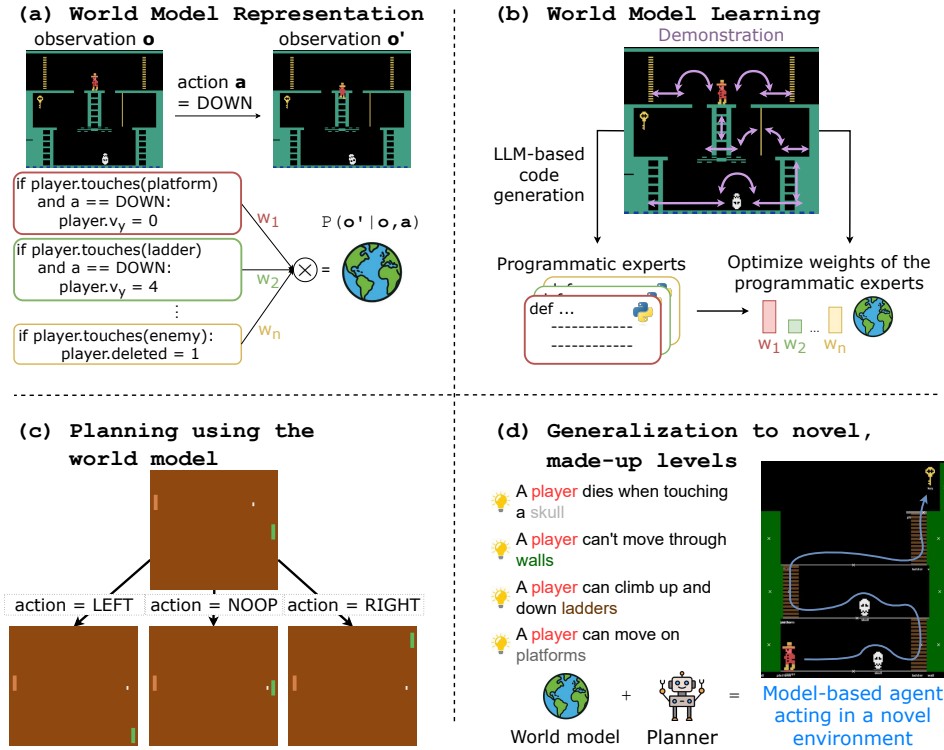

Figure 1: (a) World models predict the next state given a state-action history. We do this with a product of experts of many small programs. (b) The learner is given a short (<1 min) demonstration of gameplay as input, and uses it to synthesize initial world-model programs. These programs are refined online in later environment interactions. (c) World models support planning by imagining future states. (d) Symbolic programs encode abstract knowledge that generalizes to new game levels.

knowledge more modular, and also more learnable, because we no longer search for a single monolithic program handling everything at once. The resulting system, which we call PoE-World (**P**roduct **o**f programmatic **E**xperts), can build elaborate world models that accurately support planning and reinforcement learning (RL) from even a brief demonstration in complex Atari games, such as Montezuma's Revenge. PoE-World handles stochasticity because the product of programs is probabilistic, and, as we show, further handles partial observability. To the best of our knowledge, this is the first time a symbolic world model has been learned for environments of this complexity.

Importantly, although PoE-World models fine-grained pixel-level movement, it does *not* model pixel-level visual appearance, instead assuming symbolic observations from an object detector. Unlike model-based reinforcement learning, PoE-World does not attempt efficient exploration, but focuses on faithfully learning from a demonstrated trajectory (Figure 1b). Lastly, while the world models are ultimately used for planning (Figure 1c), PoE-World fundamentally focuses on world modeling, and not on solving the challenging computational problem of planning itself.

Despite these limitations, we view PoE-World as addressing a central learning problem: Given limited demonstrations of a new environment, quickly assemble a working world model that compositionally generalizes to new situations (Figure 1d). We highlight the following contributions:

1. The PoE-World representation and learning algorithm for symbolic world models.

2. An empirical study of PoE-World on two representative Atari games, Pong and Montezuma's Revenge, demonstrating its superior learning efficiency compared to deep RL, and improved scalability compared to state-of-the-art symbolic model-based RL: PoE-World can synthesize 4000+ line programs that generalize zero-shot to novel game levels and game variations.

3. Demonstration of how to use PoE-World's world models for planning-based decision making, and as a simulated pretraining environment for deep RL.

## 2 Background: World Model Learning

A sequential decision-making problem can be described as $(\mathcal{O}, \mathcal{A}, P, R)$ where $\mathcal{O}$ is an observation space, $\mathcal{A}$ is an action space, $P$ is an environment dynamics $P = p_{env}(o_{t+1}|o_{1:t}, a_{1:t})$ where $o \in \mathcal{O}$ and $a \in \mathcal{A}$, and $R$ is a reward function. This full-history environment formulation is mathematically equivalent to a Partially Observable Markov Decision Process (POMDP) formulation [17]. In the setting where $P$ is unknown to an agent, the agent learns by interacting with the environment and observing transitions $(o_t, a_t, o_{t+1}, r_t)$ at timestep $t$.

We focus on learning the world model $\hat{P} = p_{model}(o_{t+1}|o_{1:t}, a_{1:t})$ which approximates the true, unknown environment dynamics $P = p_{env}(o_{t+1}|o_{1:t}, a_{1:t})$ given observed trajectories $D = \{\tau_i\}_{i=1}^{n}$ where each trajectory is a sequence of observations and actions $\tau = (o_{1:T+1}, a_{1:T})$ for some $T$ ($r_{1:T}$ is also a part of a trajectory, but we drop it for simplicity). The learned world model will later be used by a model-based agent, either through lookahead planning or RL training, to act in the environment.

We treat the world modeling problem as an optimization problem, specifically empirical risk minimization:

$$p_{model}^{*} = \arg\min_{p_{model}} \sum_{(o_{1:T+1}, a_{1:T}) \in D} \sum_{t=1}^{T} \ell(p_{model}; o_{1:t+1}, a_{1:t}) \tag{1}$$

where $\ell$ is a loss function, such as the negative log likelihood function $\ell(p_{model}; o_{1:t+1}, a_{1:t}) = -\log p_{model}(o_{t+1}|o_{1:t}, a_{1:t})$.

Previous works in the model-based reinforcement learning literature [18, 19, 1, 20, 21, 22, 23] have used various deep neural network architectures, including convolutional and recurrent neural networks, transformers, and diffusion models, to define $p_{model}$ as a parametric model $p_{model} = p_{\boldsymbol{\theta}}$. Then, $\boldsymbol{\theta}$ is optimized via gradient descent. A weakness of such approaches is poor sample efficiency and generalization. For example, Diamond [22] has failure modes such as imagining that the player can walk through a wall or teleport, even after training on almost 100 hours of observations.

Other works leverage Large Language Models (LLMs) to synthesize a code world model [3, 24, 25], which can be seen as searching for programmatic $p_{model}$. These LLM-based code generation algorithms are more sample efficient and extrapolate more systematically than deep learning approaches. However, they have yet to succeed beyond simple text-based and gridworld games.

## 3 Modeling the World as a Product of Programmatic Experts (PoE-World)

To address the limitations of existing world model learning methods, we propose representing world models as exponentially-weighted **P**roducts **o**f programmatic **E**xperts (**PoE-World**), enabling sample-efficient and scalable learning of probabilistic world models that leverages LLM code generation. Figure 1 visualizes both the representation and learning algorithm discussed in this section.

### 3.1 World Model Representation: Product of Programmatic Experts (PoE-World)

World models represented as exponentially weighted **P**roducts **o**f programmatic **E**xperts (**PoE-World**) can be described mathematically as follows:

$$p_{\boldsymbol{\theta}}(o_{t+1}|o_{1:t}, a_{1:t}) \propto \prod_i p_i^{expert}(o_{t+1}|o_{1:t}, a_{1:t})^{\theta_i} \tag{2}$$

where $p_i^{expert}$ are programs and $\theta_i$ are their associated scalar weights.

This representation enables modularity and compositionality. It allows composing many small programs into a full world model to capture complicated environmental dynamics (see Figure 2). We can think of each program as an expert which expresses opinions about particular aspects of the world. For example, in the context of modeling a video game environment, one expert might encode "if a player touches a skull, then the player dies," while another encodes "if an action is LEFT when the player is on a platform, the player's x-axis velocity is -2." The former expert does not express opinions on the player's movement, while the latter does not mention conditions for player's death.

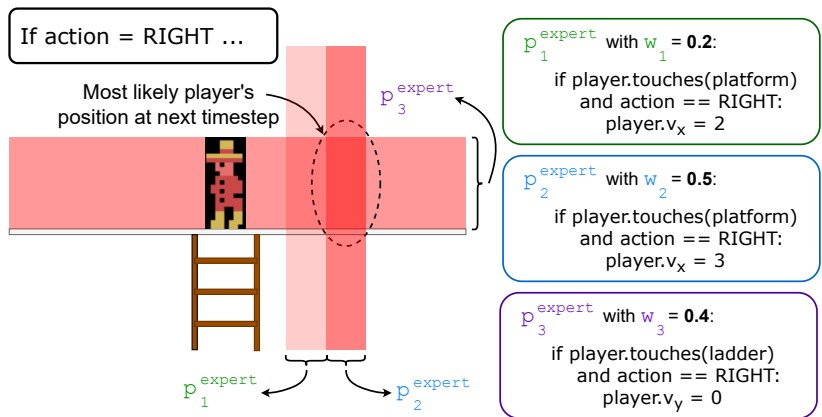

Figure 2: A "heat map" illustration of how simple Python programs are interpreted as distributions, and how they are combined into a single distribution over next-timestep object locations.

**A factored state representation yields tractable inference.** We assume an object-centric state where each object has a bounding box and velocities, each stored as a separate slot, attribute, or feature. We treat each such feature as conditionally independent, given the history so far. This independence assumption makes Equation (2) tractable, because we can compute a separate normalizing constant for each feature. Concretely, let $f$ index the different object features of the next observation $o_{t+1}$, and let $o^f$ be the indexing of $o$ with feature $f$. Then,

$$p_{\boldsymbol{\theta}}(o_{t+1}|o_{1:t}, a_{1:t}) = \prod_f \frac{1}{Z_f} \prod_i p_i^{expert}(o_{t+1}^f|o_{1:t}, a_{1:t})^{\theta_i} \quad Z_f = \sum_{o^f} \prod_i p_i^{expert}(o^f|o_{1:t}, a_{1:t})^{\theta_i}$$

**Benefits of the full-history formulation over POMDP formulation.** The full-history environment formulation is formally equivalent to a POMDP which instead compresses the history into a Markov latent state. We use the history formulation because it makes our world models more modular. Learning global latent variables would entangle all the experts, because every expert would condition on the latent state. Therefore, learning a new latent variable (e.g. "how long the player has been falling") changes the input/output space of every expert, necessitating global joint updates to the structure of every program (e.g. an expert for "is the player dead" would need to be updated). The history formulation allows independent learning of independent mechanisms.

**Hard constraints.** Atari (and the real world) is too complex to perfectly simulate with any effectively learnable program. Therefore, our probabilistic model tends to over-approximate the set of possible futures, giving fuzzy approximate predictions. For example, in Montezuma's Revenge, instead of perfectly modeling the physics of falling downward and landing on the ground, we predict a generic downward trajectory. Ideally that trajectory would perfectly enforce the constraint that the player lands flat on the ground, and never sinks into the ground, but a fuzzy stochastic expert for falling downward could violate that constraint. Therefore, to sharpen the model's outputs, we further learn a collection of hard constraints, $\{c_j\}$, where $c_j : \mathcal{O} \to \{0, 1\}$:

$$p_{\boldsymbol{\theta}}(o_{t+1}|o_{1:t}, a_{1:t}) \propto \prod_i p_i^{expert}(o_{t+1}|o_{1:t}, a_{1:t})^{\theta_i} \cdot \mathbb{1}\left[\bigvee_j c_j(o_{t+1})\right] \tag{3}$$

We further discuss hard constraints and provide concrete examples in Appendix A.1.

**Multi-timestep predictions.** We represent the programmatic experts and the world as distributions over the next-timestep observations in eq. (2). It is likewise possible to reformulate a multi-timestep expert $p^{expert}(o_{t+1:t+H}|o_{1:t}, a_{1:t})$ as a product of next-timestep experts by assuming that the predictions of the multi-step expert at different timestep are independent:

$$p^{expert}(o_{t+1:t+H}|o_{1:t}, a_{1:t}) = \prod_{k=1}^{H} p^{expert}(o_{t+k}|o_{1:t}, a_{1:t}) \tag{4}$$

## 3.2 World Model Learning: Program Synthesis and Weight Optimization

We begin with a demonstration trajectory, learn a world model, and then begin to act in the world according to that model. As the agent acts, it collects more trajectory data, which it uses to update or "debug" its model. Concretely, learning proceeds as follows:

Step 1: Synthesize programmatic experts $\{p_i^{expert}\}_{i=1}^m$ given observed trajectories $D = \{\tau_i\}_{i=1}^n$

Step 2: Fit the weights $\boldsymbol{\theta}$ of the experts according to eq. (5) with a gradient-based optimizer

Step 3: Remove the experts with weights below threshold $\delta$

Step 4: Repeat Step 1-3 every time the observed trajectories get updated

**Generating program experts.** Following previous works [3, 24, 25], we adopt Large Language Models (LLMs) as our Python program generator. We input a small batch of transitions $(o_{t:t+H+1}, a_{t:t+H})$ to the LLM prompt to produce the programmatic experts $\{p_i^{expert}\}$.

Figure 2 shows how we interpret small Python programs as distributions over the observations. While we could ask LLMs to synthesize probabilistic programs to specify the distributions, we find it much more effective to have LLMs synthesize simple, deterministic Python programs, presumably because generic Python code is far more prevalent in LLM training data. We assume that an observation is represented as a list of objects, where each object has the following attributes: x/y velocity and visibility. Then, a distribution over observations is a distribution over each object's attributes. To interpret a Python program as a distribution, we assume that all object attributes are conditionally independent given full history, as mentioned in Section 3.1, and convert all object attributes set by the program to distributions with single peaks at the given values. We then add noise to the distributions to ensure non-zero probabilities over alternative values. Any object attributes whose values are not set by the program follow uniform distributions over all possible values: consequently, experts with a single if-condition (fig. 2) yield a uniform distribution when the if-condition is not satisfied.

**Gradient-based Weights Optimization.** Once we have $\{p_i^{expert}\}_{i=1}^m$, we can perform maximum likelihood estimation to obtain $\boldsymbol{\theta}$:

$$\boldsymbol{\theta}^* = \arg\max_{\boldsymbol{\theta}} \sum_{(o_{1:T+1}, a_{1:T}) \in D} \sum_{t=1}^{T} \log p_{\boldsymbol{\theta}}(o_{t+1} | o_{1:t}, a_{1:t}) \tag{5}$$

This equation instantiates eq. (1) by letting $p_{model}$ have a parametric form $p_{model} = p_{\boldsymbol{\theta}}$ and choosing negative log likelihood as the loss function. We can use any gradient-based optimizer to optimize the weights. We use L-BFGS [26], which worked better than Adam [27] and SGD, because we have small data and few parameters.

Finally, the experts whose weights are below a threshold $\delta$ are removed from the world model. We repeat this loop every time there are new observations. Appendix A.1 contains full algorithm details.

## 3.3 World Model Usage: RL in Simulation and Planning with World Model

An important goal of world modeling is to aid decision-making. We consider two such ways of using world models. First, a world model can serve a simulator for reinforcement learning. This makes RL policy learning more sample efficient, because we can quickly learn a world model from real environment interactions, which then substitutes the actual environment. Subsequent policy learning can take place in the world model, obviating the need for further interaction with the real environment. In practice, we also continue reinforcement learning after pretraining in the world model. Formally, we learn a policy $\pi : \mathcal{O}^* \to \mathcal{A}$ that inputs an observation history and outputs an action.

Alternatively, world models can be used for lookahead planning. Given a reward function $R$, we plan for a horizon of $H$ timesteps by searching for an optimal action sequence, given our previous observations $o_{1:t}$ and actions $a_{1:t-1}$:

$$a_{t:t+H}^* = \arg\max_{a_{t:t+H}} \mathbb{E}_{p_{\boldsymbol{\theta}}(o_{t+1:t+H} | o_{1:t}, a_{1:t+H})} \left[ \sum_{k=0}^{H-1} R(o_{t+k+1}; o_{1:t+k}, a_{1:t+k}) \right]$$

where $p_{\boldsymbol{\theta}}(o_{t+1:t+H} | o_{1:t}, a_{1:t+H}) = \prod_{k=0}^{H-1} p_{\boldsymbol{\theta}}(o_{t+k+1} | o_{1:t+k}, a_{1:t+k})$.

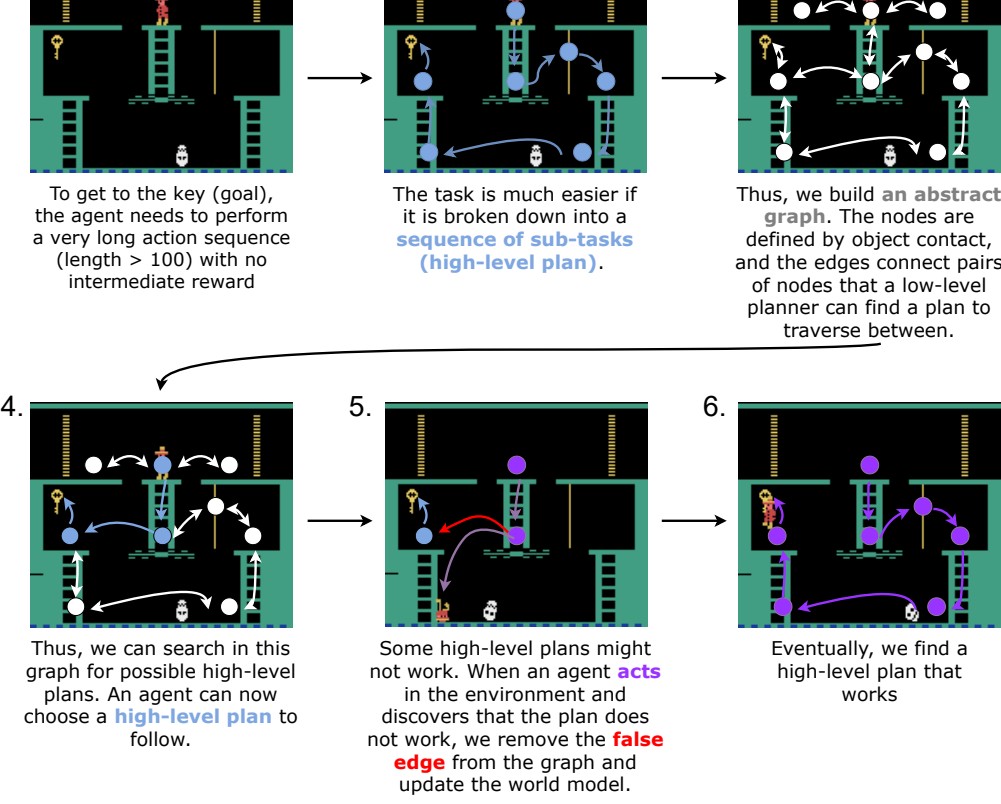

Figure 3: A sequence of illustrations that demonstrates how our hierarchical planner works.

To help our world models guide long-horizon decision-making, we implement a hierarchical planner inspired by task and motion planning (TAMP). It first plans in a high-level abstract state space defined by object contact, then lowers those plans into actual Atari button presses using the learned world model (Figure 3, Appendix A.2). The resulting agent we denote **PoE-World + Planner**.

## 4 Experimental Results

**Domains and Evaluation.** We evaluate our agent, **PoE-World + Planner**, against other methods on Atari's Pong and Montezuma's Revenge (MR), using the Arcade Learning Environment [28]. We use OCAtari [29] to parse each image frame as a list of objects, each with an object category, a bounding box, and velocities. [1] Both games are partially observed: the current state and action cannot uniquely determine the next state. A demonstration of fewer than 1000 frames is created for each game, but these demonstrations are not successful gameplays: they serve only to illustrate the core causal mechanics. In Montezuma's Revenge, our demonstration never achieves positive reward.

To test compositional extrapolation , we created alternative versions of both games, called Pong-Alt and Montezuma's Revenge-Alt (Figure 4), both of which recombine and rearrange the types of objects seen in the training demonstration. Pong-Alt increases the number of objects (3 balls and 3 enemies). Montezuma's Revenge-Alt adds more enemies (which the player has to jump over) and ladders, while changing the map to resemble the game Kangaroo (see Appendix A.3 for details). We do not provide demonstrations for these alternative versions of the games.

---

[1]OCAtari does not reliably extract objects so we have to make custom changes to OCAtari for every single Atari game we apply our method to, which makes it infeasible to run on the full Atari suite. We choose Pong and Montezuma's Revenge (arguably one of the hardest games in Atari—[30] calls it a "grand challenge") as representative games for which we manually patch OCAtari to reliably detect objects.

| Method | Score | | | |
|---|---|---|---|---|
| | Pong | Pong-Alt | MR | MR-Alt |
| Random Agent | $-20.67 \pm 0.33$ | $-20.00 \pm 0.58$ | $0.00 \pm 0.00$ | $0.00 \pm 0.00$ |
| PPO @ 100k env steps [31] | $-21.00 \pm 0.00$ | $-18.66 \pm 0.88$ | $0.00 \pm 0.00$ | $0.00 \pm 0.00$ |
| LLM as Agent (ReAct) [32] | $-20.00 \pm 0.00$ | $-20.67 \pm 0.33$ | $0.00 \pm 0.00$ | $0.00 \pm 0.00$ |
| WorldCoder + Planner [3] | $-17.00 \pm 3.00$ | $-19.00 \pm 1.00$ | $0.00 \pm 0.00$ | $0.00 \pm 0.00$ |
| PoE-World + Planner (Ours) | $\mathbf{-12.33 \pm 0.88}$ | $\mathbf{-13.67 \pm 0.67}$ | $\mathbf{100 \pm 0.00}$ | $\mathbf{100 \pm 0.00}$ |
| PPO @ 20m env steps [31] | $17.00 \pm 0.58$ | $1.33 \pm 2.03$ | $0.00 \pm 0.00$ | $0.00 \pm 0.00$ |

Table 1: Scores on Pong and Montezuma's Revenge (MR) and their alternate versions. For PoE-World and WorldCoder, brief demonstrations on Pong and MR are given to initialize the world models. Their agents then train for at most 3k steps before evaluation.

**Baselines.** **PPO** [31] is a go-to, widely-used model-free RL algorithm. It optimizes a lower bound of the policy's performance using gradient descent. **LLM as Agent (ReAct)** [32] directly uses an LLM as a policy. ReAct prompts LLMs to use extra chain-of-thought [33] "thinking" actions before selecting an action. **WorldCoder** [3] is an LLM agent that models the world as a single Python program. It uses an LLM code generation and repair algorithm called REx [34] to refine its world model to achieve high predictive accuracy on observed trajectories. More details in Appendix A.4.

**Agent Results.** Figure 5 and Table 1 show the scores of different agents on Pong, Pong-Alt, Montezuma's Revenge, and Montezuma's Revenge-Alt. The scores on Pong and Pong-Alt indicates the difference in points achieved by the player and the enemies when the game ends at 21 points. The scores on Montezuma's Revenge and Montezuma's Revenge-Alt become positive if and only if the agent succeeds in collecting the key. As shown in Table 1, our agent, **PoE-World + Planner**, performs best across all environments in the low-data regime, particularly when PPO baseline is allowed to have $100,000$ training environment interactions, the standard budget for sample-efficient agents on Atari 2600 [20]. In Figure 5, we keep training PPO for more steps, finding that it takes over a million steps for PPO to surpass PoE-World + Planner. Moreover, PPO with 20M training steps never achieves positive score on Montezuma's Revenge. PoE-World + Planner is the only method that manages to obtain positive reward on Montezuma's Revenge in both base and alternative versions.

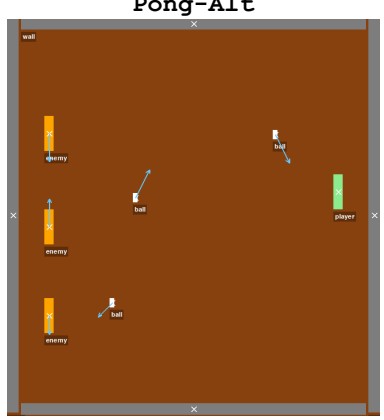
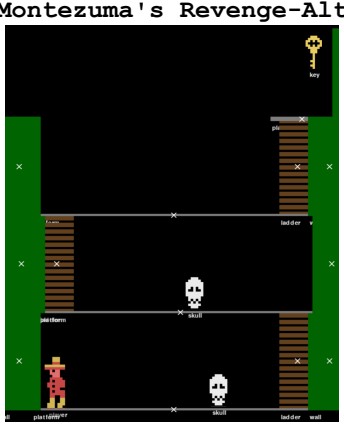

Figure 4: Screenshots of the alternative environments. Pong-Alt's objective is to hit three balls past three enemies to score points, while Montezuma's Revenge-Alt requires players to avoid moving skulls by jumping over them and climbing up ladders to reach and collect the key.

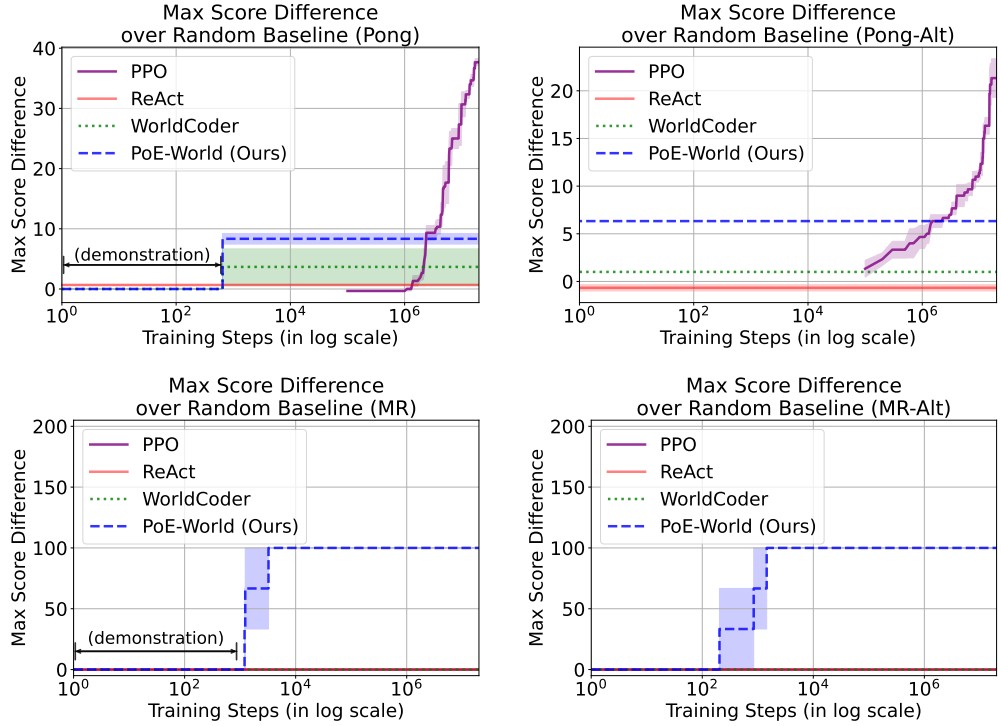

Figure 5: Maximum score differences over random baseline achieved at different number of training steps (in log scale). The text "(demonstration)" annotates the length of demonstration used to initialize WorldCoder and PoE-World, which is not displayed for the Alt games since we use the world models initialized by the demonstrations of the corresponding base games.

**Instead of planning, can we use PoE-World to learn a policy?** Training a policy avoids the test-time compute cost of planning. In Figure 6, we show that pre-training a policy inside PoE-World's world model accelerates policy learning: we first run PPO "in simulation" (in our world model), and then fine-tune in the actual Atari environment. The fine-tuned PPO achieves significantly higher score than vanilla PPO at most training steps, and while the vanilla PPO takes 1M training steps to do better than a random agent, the fine-tuned PPO takes only 200k training steps. Asymptotically the pretrained and randomly-initialized policies converge to the same value: World-model pretraining is effectively a way of warm-starting policy training.

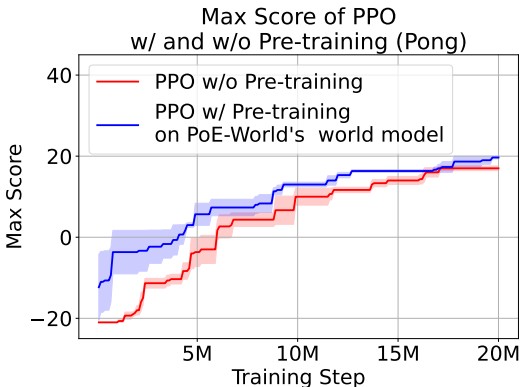

Figure 6: PPO with and without pre-training within our world model.

**Next Observation Prediction Results.** Table 2 and Table 3 show next observation and next observation's object attributes prediction accuracies under the symbolic world modeling approaches, respectively. PoE-World outperforms the baselines on most settings, except for the test observation (random frames) for Pong where all methods perform similarly because a random agent rarely succeeds in hitting the ball, and so knowledge of game mechanics is not comprehensively tested.

**The role of hard constraints.** We investigate the role of hard constraints in our world model representation in Table 4. In practice, hard constraints act to rule out "physically impossible" scenes. PoE-World ended up enforcing constraints just for MR, since most possible scenes in Pong are already physically possible. As shown in Table 4, removing hard constraints causes the agent's performance

| Method | Next Observation Prediction Accuracy | | | | | |
| --- | --- | --- | --- | --- | --- | --- |
| | Train | | Test (1000 Random Frames) | | | |
| | Pong | MR | Pong | Pong-Alt | MR | MR-Alt |
| WorldCoder | $0.33 \pm 0.01$ | $0.36 \pm 0.02$ | $0.06 \pm 0.01$ | $0.01 \pm 0.01$ | $0.10 \pm 0.01$ | $0.12 \pm 0.03$ |
| PoE-World (ours) | $\mathbf{0.51 \pm 0.01}$ | $\mathbf{0.75 \pm 0.00}$ | $0.06 \pm 0.01$ | $\mathbf{0.03 \pm 0.00}$ | $\mathbf{0.31 \pm 0.00}$ | $\mathbf{0.43 \pm 0.00}$ |

Table 2: Next observation prediction accuracy on the training demonstration and 1000 random frames.

| Method | Next Observation's Object Attributes Prediction Accuracy | | | | | |
| --- | --- | --- | --- | --- | --- | --- |
| | Train | | Test (1000 Random Frames) | | | |
| | Pong | MR | Pong | Pong-Alt | MR | MR-Alt |
| WorldCoder | $0.82 \pm 0.00$ | $0.75 \pm 0.01$ | $0.66 \pm 0.01$ | $0.67 \pm 0.02$ | $0.58 \pm 0.00$ | $0.65 \pm 0.02$ |
| PoE-World (ours) | $\mathbf{0.88 \pm 0.00}$ | $\mathbf{0.93 \pm 0.00}$ | $0.66 \pm 0.00$ | $\mathbf{0.76 \pm 0.00}$ | $\mathbf{0.76 \pm 0.00}$ | $\mathbf{0.83 \pm 0.00}$ |

Table 3: Next observation's object attributes prediction accuracies on the training demonstration and 1000 random frames. Non-moving objects, such as walls, platform, etc. are excluded.

| Method | Planning successes | | Next Observation Prediction Accuracy (Test) | |
| --- | --- | --- | --- | --- |
| | MR | MR-Alt | MR | MR-Alt |
| PoE-World + Planner w/o Hard Constraints | 2/9 | 2/9 | $0.30 \pm 0.01$ | $0.43 \pm 0.00$ |
| PoE-World + Planner | $\mathbf{5/9}$ | $\mathbf{4/9}$ | $0.31 \pm 0.00$ | $0.43 \pm 0.00$ |

Table 4: Planning successes (out of 9 tries), where the goal is for the player to grab the key, and next observation prediction accuracies on 1000 random test frames of our agent with and without hard constraints on Montezuma's Revenge (MR) and its alternate version (MR-Alt).

to drop on both the base and alternative versions of Montezuma's Revenge. Interestingly, we found no significant changes in next observation prediction accuracies. We hypothesize that hard constraints do not necessarily turn bad predictions into good ones. Instead, they perform damage control—they refine poor predictions just enough to make them usable for long-horizon planning.

**Qualitative difference between WorldCoder's and PoE-World's world models.** World models produced by PoE-World consists of 4000+ lines of code for Montezuma's Revenge compared to WorldCoder's less than 100 lines. This difference is reflected in their ability to capture the underlying causal laws: PoE-World models accurately represent important causal laws like character movement constraints with respect to platforms and ladders, while WorldCoder fails to model these mechanics, predicting that the player can "fly" around the map without any constraints. Moreover, WorldCoder models tend to hallucinate, e.g., imagining nonexistent bullet firing abilities in Montezuma's Revenge, potentially because there is no granular way to downweigh buggy parts of a world model. PoE-World, in contrast, prunes irrelevant experts with low weights, yielding more precise world models.

## 5 Related Work

**World models as programs** has been explored in several recent works that motivate PoE-World. Similar to our work, WorldCoder [3] and CodeWorldModels [24] use LLMs to write a Python transition function. Unlike our work, they learn a single monolithic program, which severely limits scalability: our world models have an order of magnitude more code needed to model complex environments, as well as handle partial observability and non-determinism. Recent works have also explored synthesizing high-level abstract world models as programs [35, 36] to support robotic planning by integrating visual perception with symbolic reasoning. They could synergize with our work, as we have focused on learning low-level world models describing the motion of objects.

Earlier works [37, 38] learn world models in restricted (non-Turing complete) domain specific programming languages, performing well on benchmarks co-designed with their Domain Specific Language. Even earlier, Schema Networks learn a conceptually related factor-graph world model [39]. AIXI [40] is a theoretical model of reinforcement learning which considers all possible Turing

machines, mathematically related to all these works. Interestingly, AIXI also works with a history space as a conceptually elegant way of modeling partial observability. Other studies generate programs for world models primarily from natural language instructions, rather than from example interactions with the environment [41, 42, 43, 44].

**Hierarchical Planning.** Our approach to segmenting complex continuous tasks into symbolic states is inspired by hierarchical [45] and task-and motion [46, 47] planning. Hierarchical representations in Reinforcement Learning (RL) are often expressed as options—temporal abstractions that allow agents to reason at multiple time scales by grouping sequences of primitive actions [48]. Options can be manually specified by design or learned from experience [49]. Task and Motion Planning (TAMP) automatically discover symbolic states and action abstractions that enable generalizable symbolic plans grounded in continuous motion, however the applications of TAMP to Atari-style domains are still lacking due to complex dynamics. Similar to recent work [29, 50], we define a high-level state space in terms of object contact relations—which is key to supporting symbolic abstraction.

**Alignment with human cognition.** Our system, composed of simpler, specialized programmatic experts that give rise to complex behavior, is inspired by a view of mind as a community of interacting agents —- a recurring theme in philosophy and cognitive science [5, 6, 52, 7]. Our modeling approach to modeling objects and actions aligns with empirical studies of event segmentation [53, 54] and hierarchical planning [55, 56]. Hierarchical state-spaces based on motion cues, such as ours, predict how people draw [13] and interpret social interactions [57], attesting to the cognitive alignment of our approach. Likewise, our programmatic representations of actions align with studies that demonstrate human concept learning to be akin to mental programs [58, 59, 11, 12], which recent work models by LLM-based program synthesis.

## 6   Discussion and Limitations

**Compositionality.**   Recombining pieces of knowledge to generalize and extrapolate is a core feature of symbolic systems and, arguably, also of human cognition, ranging from natural language to abstract reasoning to everyday thought [60, 52, 61, 62]. Our approach is in this spirit. It factors its knowledge into small experts whose predictions can combine to extrapolate to scenes with more objects recomposed into new arrangements. More formally, our approach generalizes to novel "entity compositions" and "relational compositions," terms used by [63]. It should be noted, however, that our compositional factoring is orthogonal to our use of symbolic code: Programs can be monolithic [3], and neural nets can be factored [64]. Nonetheless, compositionality proved critical to scalably learning the underlying symbolic program. Synthesizing a single monolithic program is, in our view, intractable not just for the real world, but even for Atari.

**Limitations.**   We make important assumptions, and only address part of the full model-based reinforcement learning problem. Symbolic programs expect symbolic inputs: We do not learn straight from pixels. RL involves exploration, decision-making, and reward function learning, but we do not address those problems here. However, we speculate our approach could unlock better methods for exploration: A program-structured world model exposes an interpretable interface for describing beliefs about how the world works, and efficiently exploring the world is analogous to testing the program that encodes the world model. Therefore ideas from software testing and program analysis could, in theory, be brought to bear, enabling new approaches to model-based exploration.

## Acknowledgments

We thank Edward Gu and Taha Jafry for their contributions to earlier versions of this project. We are also grateful to Atharv Sonwane, Yilun Du, and Tom Silver for their valuable discussions and feedback on the paper, and we appreciate the OCAtari team's assistance and support with the OCAtari package. This work was supported by an NSF CAREER grant.

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

# A   Appendix

## A.1   PoE-World algorithm details

As mentioned in Section 3.2, our learning algorithm PoE-World alternates between two main steps: programmatic expert synthesis and gradient-based weight optimization. Given a trajectory, we do batch processing of size 10—we first learn $p_\theta(o_{11}|o_{1:10}, a_{1:10})$, then $p_\theta(o_{21}|o_{1:20}, a_{1:20})$, and so on.

**Programmatic expert synthesis.**   To synthesize programmatic experts, we first need to turn the observations into natural language observations that can be input into LLMs. A sequence of consecutive observation transitions $(o_{t:t+H+1}, a_{t:t+H})$ can be transformed into a text showing the first input object list, the list of actions, and the changes to the input object list for the following timesteps, all in natural language. Table 5 shows an example text representation.

We note that in the text representation, only changes to a single object type (which is 'player' in Table 5) is displayed at a time. This choice is to enforce modularity between the object types: each experts put non-uniform distribution only on a single attribute of objects with a specific object type. This means Equation (2) can be further rewritten as:

$$p_\theta(o_{t+1}|o_{1:t}, a_{1:t}) \propto \prod_{obj-type} \prod_i p_i^{obj-type\_expert}(o_{t+1}|o_{1:t}, a_{1:t})^{\theta_i} \tag{6}$$

where $\{p_i^{obj-type\_expert}\}$ are the experts associated with each object type.

We then implement multiple (10 in total) LLM-based synthesis modules. For each object type, these modules take as input a sequence of consecutive observation transitions and output a set of programmatic experts. The output programmatic experts are pooled into a single set once all modules finish running. Having multiple synthesis modules allows each module to focus on different aspects of the environment, e.g., how objects move passively, how object moves when interacts with objects, how objects get created and deleted, etc. We describe one of our modules, ActionSynthesizer, below.

ActionSynthesizer focuses on synthesizing experts that explain how each action affects objects when the objects are interacting (touching) other objects (see example experts of this kind in Figure 2). It takes in just a single transition $(o_{t:t+1}, a_t)$. It turns this transition into a text representation as discussed above (Table 5). Then, it prompts a LLM to output causal explanations for the object changes (see the prompt Table 6). An example causal explanation is "the player objects that touch an ladder object set their y-axis velocity to -4". With a set of causal explanations, we prompt a LLM to turn each of them into a program (see the prompt Table 7 and Table 8).

Other modules follow this template of first prompting a LLM for natural language causal explanations and then prompting a LLM to turn the explanations into programs. We refer the reader to our code at `https://github.com/topwasu/poe-world` for the implementations of other LLM-based module.

The programs synthesized by LLMs use our manually-written helper classes: `Obj`, `ObjList`, `RandomValues`, and `SeqValues`. Their docstrings are passed to the prompt. `Obj` provides a method `Obj.touches` which is the function we manually build in to help determine object contact. `Obj`

---

```
Example input list of objects:
player object (id = 0) with x-axis velocity = +4,
Interaction -- player object (id = 0) is touching ladder object (id = 2),
Interaction -- player object (id = 0) is touching unknown object (id = 4),

Example list of actions:
NOOP, NOOP, RIGHT

Example output list of object changes:
- The player object (id = 0) sets x-axis velocity to [+0, +0, +2]
```

---

Table 5: Example text representation of a sequence of consecutive transitions.

contains object attributes: object category, x/y position, x/y velocities. It also contains properties including `center_x`, `center_y`, `left_side`, `right_side`, which are calculated based on the position and velocities, and their setter methods actually modify the velocities under the hood. `ObjList` is a class that represents an object list. It has a method `ObjList.get_objs_by_obj_type` and `ObjList.create_object`. Lastly, `RandomValues` and `SeqValues` are the classes used to mark values set by a LLM. The purpose of these two classes is further discussed below.

After the programmatic experts are synthesized (see examples in Table 9), we interpret them as distributions as discussed in Section 3.2. We tell LLMs in the prompt to set attribute values as instances of `RandomValues`, as opposed to integers. This marks the attributes whose values changed by a LLM so that we can write an program-to-distribution interpreter that puts a single-peak distribution on the attribute whose value is set and uniform distributions on all other attributes. `SeqValues` is similar to `RandomValues`, but is used in multi-timestep predictions scenario, as discussed in Section 3.1.

The LLM used in the steps above is gpt-4o-2024-08-06. We implemented a disk cache for the LLM responses to avoid paying multiple times for the same prompts and seeds.

**Hard constraints.** The hard constraints (Section 3.1) are also synthesized similar to how we synthesize the programmatic experts. Table 10 shows the constraints learned for Montezuma's Revenge. In Equation (3), we choose to use a disjunction rather than a conjunction because the physics in video games can be peculiar and unrealistic—a player might have their body overlap with a platform when climbing down a ladder attached to that platform. In the real world, we believe a conjunction would be a better choice.

**Gradient-based weight optimization.** Once we have the expert distributions $\{p_i^{expert}\}$, we can optimize their weights $\{w_i\}$ according to Equation (5). We use the L-BFGS optimizer [26] implemented in PyTorch [76] with strong Wolfe line search, learning rate = 1, number of epochs = 4, and without mini-batching. We also include a L1 regularization loss with weight = 1 so that the weights do not get too big.

We note that the weight optimization is done without taking into account the hard constraints since we would like to use gradient-based approaches.

We prune programs with weights lower than $\delta = 0.01$ after the weight optimization is done, and we prune constraints that contradict with the observations or explain less than 1% of the observations.

## A.2 Planner

In a game like Montezuma's Revenge, planning in the actual, low-level action space is hard because the number of actions required to get to the first positive reward is very high—at least 100. This means the size of search space is $8^{100}$ as there are 8 possible actions: NOOP, UP, DOWN, LEFT, RIGHT, FIRE, LEFTFIRE, RIGHTFIRE.

Thus, inspired by task and motion planning (TAMP), our hierarchical planner interleaves planning in the low-level motion action space with planning in a high-level abstract action space. It learns an abstract graph where the nodes are abstract states defined by object contact, and the edges represent whether the world model believes the player can traverse between the two nodes. Then, we search for a path in the abstract graph that takes the player to the goal. In Atari, a goal is usually for a player to touch a goal object (key, ball, platforms, etc.). The discovered path in the abstract graph is a high-level plan—a sequence of subgoals—for a low-level planning agent. A low-level planning agent performs online planning to choose an action and then execute it in the environment.

The planning algorithm can be described step-by-step as follows:

Step 1: Learn an abstract graph by running a low-level motion planner in simulation on all pairs of nodes. A (ordered) pair of nodes has an edge between them if we can find at least one low-level plan to traverse between them.

Step 2: Search for a path in the abstract graph. We use breadth-first search (BFS) here to find a path to the goal with the shortest length. The path is our high-level plan. If there is no path to the goal, go back to step 1.

Step 3: Attempt to follow the high-level plan, completing each subgoal in order, with a low-level planning agent.

Step 4: If there is a "false" edge, update the world model, remove that edge from the graph, then go back to step 2. Otherwise, the algorithm stops, and the agent has achieved the goal.

Figure 3 shows simplified illustrations of how our hierarchical planner works.

We now discuss the implementation of our low-level motion planner:

**Low-level motion planner.**  We implemented two low-level motion planners. The first is a variant of Monte Carlo Tree Search (MCTS) [77, 78]. It follows the same set of procedures as vanilla MCTS with two differences: first, similar to the MCTS algorithm used in [3], it approximates the value of a node using a heuristic function instead of doing a random rollout in the simulate step. The heuristic function is the Manhattan distance between the current position of the player object and the position of a goal object. We find that this function is a good estimate of how good an observation is when trying to achieve a goal. Second, the value of a node is updated as the maximum value of its children nodes, instead of the expected value, in the backpropagation step. Intuitively, using the maximum value encourages the planner to be more optimistic. The exploration parameter for MCTS is initially equal to 1 for Montezuma's Revenge and Montezuma's Revenge-Alt and 10 for Pong and Pong-Alt, and it increases by 10 times every 1000 iterations of MCTS.

We also implement "sticky actions" by extending the action space that MCTS searches on so that it includes repeated sequences of primitive actions with lengths 1, 4, and 8. Thus, the extended action space has $3n$ actions where $n$ is the number of primitive actions. We include these repeated action chunks in our action space to make planning easier. Playing Atari games rarely requires players to change actions at every timestep, so repeated action chunks can be helpful.

The second low-level motion planner is a greedy search with the same heuristic function used in MCTS. At each iteration, it greedily finds the best repeated action chunk of length 8 and includes it in the plan. It backtracks if the current state leads to death no matter which action chunk of length 8 the planner chooses. It returns a plan when the player achieves the goal (touches the goal object).

Pong, Pong-Alt, and Montezuma's Revenge-Alt agents only use the greedy planner, while Montezuma's Revenge uses both: it first tries to find a plan with MCTS for 4000 iterations and falls back to the greedy planner if MCTS fails.

As discussed earlier, the low-level motion planner is used in two steps: to build the abstract graph by finding a plan to traverse between two abstract nodes in the world model, and to help inform a low-level planning agent that is trying to complete a subgoal.

**Low-level planning agent.**  The agent performs online planning: it uses the low-level planner to find a plan to the goal, and then it takes a sequence of actions and replans. The agent replans when the ccurent plan no longer takes the agent to the goal, so the agent may take several actions before replanning. We further optimize the agent by letting it replan only 40% of the times when the current plan no longer works in Montezuma's Revenge.

Because of this introduced stochasticity, however, we find that our whole planning pipeline can give different scores in different runs even with the same initial world model, so we treat running the hierarchical planner multiple times with the same initial world model as part of training, and we run the planning algorithm 3 times on the same initial world model to get results on Montezuma's Revenge and Montezuma's Revenge-Alt. Our work focuses on world modeling, and we leave it to future work to increase the efficiency and performance of the hierarchical planner.

### A.3  Domain details

We evaluate our methods on Atari's Pong and Montezuma's Revenge using the Arcade Learning Environment (ALE). The frameskip parameter is set to 3 for both games. We use OCAtari to parse each image frame as a list of objects, each with an object category, a bounding box, and velocities. OCAtari reverses-engineers the RAM values of Atari games to get the bounding boxes of each object.

We modify OCAtari to fix a number of issues in its handling of Pong and Montezuma's Revenge. This includes fixing bugs in object detection so that it fully detects every object in play, correcting bugs in the the bounding boxes so that object interactions correctly correspond to when object bounding boxes touch/overlap, etc. We refer the reader to our code implementation `https://github.com/topwasu/poe-world` for full details.

Details on how we create Pong-Alt and Montezuma's Revenge-Alt are below:

**Pong-Alt** is created by layering three Pong environments. We sync the player's location in all three environments, so that it appears as if we have only one paddle. The enemies and balls, on the other hand, are all at different locations. We end up with one player, three enemies, and three balls.

**Montezuma's Revenge-Alt** is created by stacking the lower section of the first room in the original Montezuma's Revenge to make three platforms of different heights, connected by stairs. We stack up three lower sections of three different Montezuma's Revenge environments. The first and the third section in the stack are flipped horizontally so that the stairs are on different sides of the room, requiring the player to jump over the skulls to reach the stairs.

ALE code uses GPL-2.0 license, and OCAtari code uses MIT license.

## A.4 Baseline details

**PPO.** PPO uses frame stacking = 4. We use the same hyperparameters as the PPO paper [31] and OCAtari [29]. We use the stable-baselines3 implementation of PPO with MLP backbone ('MlpPolicy') [79]. For the Alt environments, we take the PPO model pretrained 20M steps on the base environments and finetune it on the Alt environments. This pretraining process is done for fiar comparison with our method, since our method assumes demonstrations from the base environments when evaluating on the corresponding Alt environments.

**ReAct.** We write prompts that would alternate between thinking and taking actions, one for Pong and Pong-Alt and another for Montezuma's Revenge and Montezuma's Revenge-Alt. The frame observation is transformed into text where we provide each object's x and y position and a list of all pairwise object interactions. This text observation is input as part of the prompt. In the prompt, we only include the 4 most recent observations (along with the 4 most recent thinking actions and 4 most recent taken actions) as each observation is quite long in text.

**WorldCoder.** We use the official WorldCoder implementation [3] but replace the existing prompts with new ones that are tailored towards our text representation of object-centric Atari frames. The prompts can be found in our codebase `https://github.com/topwasu/poe-world`. The instantiation of WorldCoder in [3] actually has its own planner implemented, but for fair comparison with our method, we use only the world modeling part of that system and combine it with our own planner so that the planner is the same for both WorldCoder and our method. We choose to use our own planner instead of theirs since ours is hierarchical.

WorldCoder code uses MIT license.

## A.5 Compute resources and execution time

**Compute Resources.** For the world modeling part, our experiments are run on 4 CPUs (Cascade-Lake, IceLake, or SaphireRapids) with 64 GB memory. PoE-World and WorldCoder uses a budget of $20 worth of OpenAI credit per run. For the planner, we also run it mostly on 4 CPUs, but for the part where we need to build an abstract graph by running many low-level planners, we parallelize it on multuple compute jobs on a job scheduling cluster, so we might be using 100 CPUs at a time.

**Execution time.** PoE-World alone without the planner tends to take around 8 hours to run (this includes the time we need to wait for OpenAI LLM requests). The planner running time varies, but most runs finish under 24 hours.

```
I'll give you an input list of objects and an output list of object
changes, and I want you to list 4 possible reasons for the effects

Here's an example with player objects:
Example input list of objects:
player object (id = 0) with x-axis velocity = +0 and y-axis velocity +2,
Interaction -- player object (id = 0) is touching ladder object (id = 2),
Interaction -- player object (id = 0) is touching unknown object (id = 4),

Example output list of object changes:
- The player object (id = 0) sets x-axis velocity to +0
- The player object (id = 0) sets y-axis velocity to -4

Example reasons:
1. The player objects that touch an unknown object set their x-axis
velocity to +0
2. The player objects that touch an unknown object set their y-axis
velocity to -4
3. The player objects that touch an ladder object set their x-axis velocity
to +0
4. The player objects that touch an ladder object set their y-axis velocity
to -4

Please output a list of 4 reasons of the {obj_type} objects for the
following input and output list of objects.

Input list of objects:
{input}

Output list of object changes:
{effects}

Please follow these rules for your output:
1. make sure each reason only talks about one object change
2. do not talk about IDs
```

Table 6: A prompt in ActionSynthesizer used to get LLM to provide causal explanations of the object changes.

We observe that the possible effects of {action} on {obj_type} objects include {obs_lst_txt}

We want to synthesize python functions that implements these effects. The format of the functions should be

```
def alter_{obj_type}_objects(obj_list: ObjList, action: str) -> ObjList:
    {obj_type}_objs = obj_list.get_objs_by_obj_type('{obj_type}') # get all Obj of obj_type '{obj_type}'
    for {obj_type}_obj in {obj_type}_objs: # {obj_type}_obj is of type Obj
        # You can assume {obj_type}_obj.velocity_x and {obj_type}_obj.velocity_y are integers.
        pass
    return obj_list
```

And here are the docstrings for relevant classes:

```
class RandomValues:
    Use this class to express the possibility of random values. Example x = RandomValues([x + 2, x - 2])

    Attributes:
        values (list[ints]): list of possible values

    Methods:
        __init__(values):
            Initialize an instance

class Obj:
    Attributes:
        id (int): id of the object
        obj_type (string): type of the object
        velocity_x (int | RandomValues): x-axis velocity of the object
        velocity_y (int | RandomValues): y-axis velocity of the object
        deleted (int | RandomValues): whether this object gets deleted (1 if it does and 0 if it does not)

    Methods:
        touches(obj: Obj) -> bool:
            Returns whether this Obj is touching the input obj (True/False)

class ObjList:
    Attributes:
        objs (list of Obj)

    Methods:
        get_objs_by_obj_type(obj_type: str) -> list[Obj]:
            Returns list of objects with the input obj_type

        create_object(obj_type: str, x: int, y: int) -> ObjList:
            Returns a new instance of ObjList with the new object (with obj_type, x, y) added
```

Table 7: First half of a prompt in ActionSynthesizer used to turn the natural language causal explanations into programs.

```
Please output {n} different alter_{obj_type}_objects functions that
explains each of the {n} possible effects of action '{action}' following
these rules:
1. Each function should make changes to one attribute -- this could be
the x-axis position, y-axis position, creation of object, or deletion of
object.
2. Always use RandomValues to set attribute values. If there are
conflicting changes to an attribute, instantiate RandomValues with a list
of all possible values for that attribute.
3. Use Obj.touches to check for interactions.
4. Avoid setting each attribute value for each {obj_type} object more than
once. For example, use 'break' inside a nested loop.
5. You can assume the velocities of input objects are integers.
6. Please use if-condition to indicate that the effects only happen because
of action '{action}'
Format the output as a numbered list.
```

Table 8: Second half of a prompt in ActionSynthesizer used to turn the natural language causal explanations into programs.

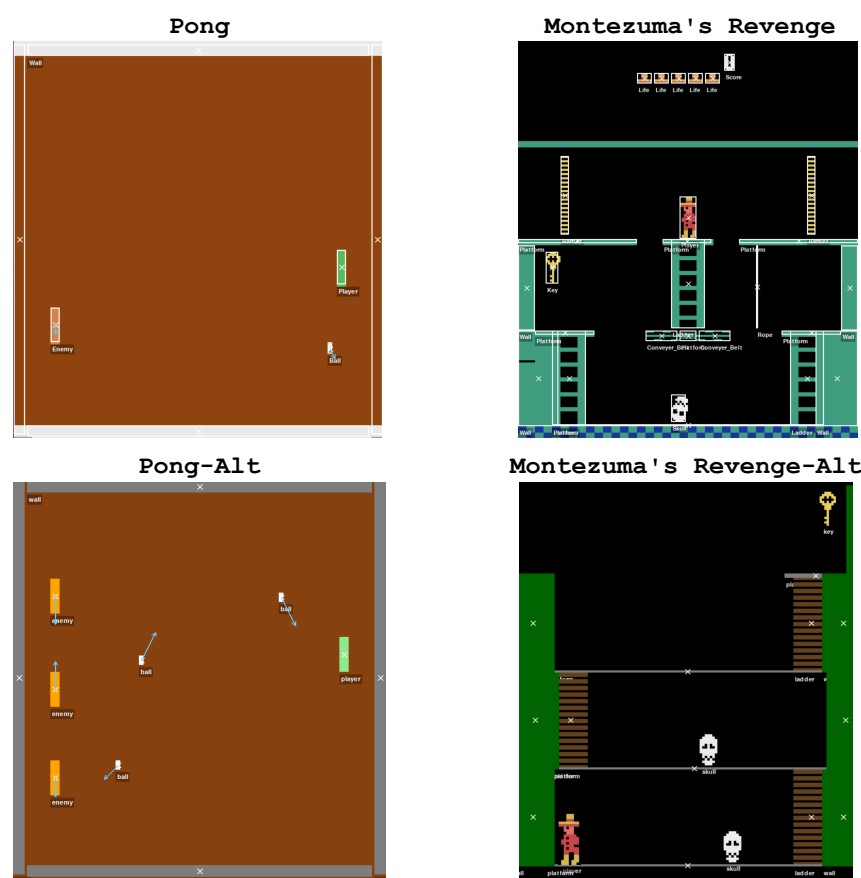

Figure 7: Screenshots of both base and alternative environments for Pong and Montezuma's Revenge.

```
# Example program 1
def alter_player_objects(obj_list: ObjList, action: str, touch_side=3,
touch_percent=0.6) -> ObjList:
    if action == 'NOOP':
        player_objs = obj_list.get_objs_by_obj_type('player')
        conveyer_belts = obj_list.get_objs_by_obj_type('conveyer_belt')
        for player_obj in player_objs:
            for conveyer_belt in conveyer_belts:
                if player_obj.touches(conveyer_belt, touch_side,
touch_percent):
                    player_obj.velocity_x = RandomValues([-1])
                    break
    return obj_list

# Example program 2
def alter_player_objects(obj_list: ObjList, action: str, touch_side=3,
touch_percent=1.0) -> ObjList:
    if action == 'FIRE':
        player_objs = obj_list.get_objs_by_obj_type('player')
        platform_objs = obj_list.get_objs_by_obj_type('platform')
        for player_obj in player_objs:
            for platform_obj in platform_objs:
                if player_obj.touches(platform_obj, touch_side,
touch_percent):
                    player_obj.velocity_y = RandomValues([-6])
                    break
    return obj_list

# Example program 3
def alter_player_objects(obj_list: ObjList, action: str, touch_side=3,
touch_percent=0.3) -> ObjList:
    if action == 'RIGHTFIRE':
        player_objs = obj_list.get_objs_by_obj_type('player')
        platform_objs = obj_list.get_objs_by_obj_type('platform')
        for player_obj in player_objs:
            for platform_obj in platform_objs:
                if player_obj.touches(platform_obj, touch_side,
touch_percent):
                    player_obj.velocity_y = SeqValues([-6, -7, -4, 0, 2, 6,
9])
                    break
    return obj_list
```

Table 9: Example synthesized programmatic experts.

```
# Constraint 1: Player's body must align with the center of the rope
def c1(obj_list: ObjList, _, touch_side=2, touch_percent=0.3) -> tuple:
    touch_ids, satisfied_ids = [], []
    player_objs = obj_list.get_objs_by_obj_type('player')  # get all Obj of type
'player'
    rope_objs = obj_list.get_objs_by_obj_type('rope')  # get all Obj of type 'rope'
    for player_obj in player_objs:  # player_obj is of type Obj
        for rope_obj in rope_objs:  # rope_obj is of type Obj
            if player_obj.touches(rope_obj, touch_side, touch_percent):
                touch_ids.append(player_obj.id)
                if player_obj.center_x == rope_obj.center_x:
                    satisfied_ids.append(player_obj.id)
    return touch_ids, satisfied_ids

# Constraint 2: Player's feet must align with the top of the conveyer belt
def c2(obj_list: ObjList, _, touch_side=3, touch_percent=0.1) -> ObjList:
    touch_ids, satisfied_ids = [], []
    player_objs = obj_list.get_objs_by_obj_type('player')  # get all Obj of type
'player'
    conveyer_belt_objs = obj_list.get_objs_by_obj_type('conveyer_belt')  # get all
Obj of type 'conveyer_belt'
    for player_obj in player_objs:  # player_obj is of type Obj
        for conveyer_belt_obj in conveyer_belt_objs:  # conveyer_belt_obj is of type
Obj
            if player_obj.touches(conveyer_belt_obj, touch_side, touch_percent):
                touch_ids.append(conveyer_belt_obj.id)
                if player_obj.bottom_side == conveyer_belt_obj.top_side:
                    satisfied_ids.append(conveyer_belt_obj.id)
    return touch_ids, satisfied_ids

# Constraint 3: Player's feet must align with the top of the platform
def c3(obj_list: ObjList, _, touch_side=3, touch_percent=0.5) -> ObjList:
    touch_ids, satisfied_ids = [], []
    player_objs = obj_list.get_objs_by_obj_type('player')  # get all Obj of type
'player'
    platform_objs = obj_list.get_objs_by_obj_type('platform')  # get all Obj of type
'platform'
    for player_obj in player_objs:  # player_obj is of type Obj
        for platform_obj in platform_objs:  # platform_obj is of type Obj
            if player_obj.touches(platform_obj, touch_side, touch_percent):
                touch_ids.append(platform_obj.id)
                if player_obj.bottom_side == platform_obj.top_side:
                    satisfied_ids.append(platform_obj.id)
    return touch_ids, satisfied_ids

# Constraint 4: Player's body must align with the center of the ladder
def c4(obj_list: ObjList, _, touch_side=3, touch_percent=1.0) -> ObjList:
    touch_ids, satisfied_ids = [], []
    player_objs = obj_list.get_objs_by_obj_type('player')  # get all Obj of type
'player'
    ladder_objs = obj_list.get_objs_by_obj_type('ladder')  # get all Obj of type
'ladder'
    for player_obj in player_objs:  # player_obj is of type Obj
        for ladder_obj in ladder_objs:  # ladder_obj is of type Obj
            if player_obj.touches(ladder_obj, touch_side, touch_percent):
                touch_ids.append(ladder_obj.id)
                if player_obj.center_x == ladder_obj.center_x:
                    satisfied_ids.append(ladder_obj.id)
    return touch_ids, satisfied_ids
```

Table 10: The collection of constraints synthesized for Montezuma's Revenge. Function names have been shortened to save space.

