# OpenReview forum: "PoE-World: Compositional World Modeling with Products of Programmatic Experts"
_NeurIPS.cc/2025/Conference — NeurIPS 2025 spotlight_

### Official Review · Reviewer_LjAF · 2025-06-27

**Clarity:** 2
**Significance:** 2
**Originality:** 3
**Rating:** 4
**Confidence:** 4

**Summary:**

The authors propose a program-synthesis approach to building a world model (i.e. transition function). Instead of attempting to write one large program to capture all the rules of the environment under consideration, they divide up the problem by having many "experts" (i.e. smaller programs) together define a distribution over the next state. They evaluate their approach on Atari's Pong and Montezuma's Revenge by looking at performance in these environments as well as next observation and next observation object attribute accuracies. They find the proposed approach to generally outperform all baselines, at least in the limited interaction setting (i.e. 100k env steps).

**Questions:**

1. In the definition of $Z_f$ in the equation after line 105, is it tractable to iterate over $o^f$? I guess the authors are using some discretized values with some range for velocity etc? And if not, how are continuous values handled?
2. On line 132, how expensive is it to repeat step 1 - 3 every time the observed trajectories get updated?
3. Line 140: "x/y velocity and visibility". Do the observations not include bounding boxes / positions of objects? And what does visibility mean?
4. In Table 2 and 3, why are the prediction accuracies typically better on the alt environments? This feels a bit counterintuitive.
5. On line 226, are the 4000+ lines of code "active" or is most of it pruned? And also, how many experts does this typically consist of? It would be great if the authors could give some insight / statistics like this on the resulting world model programs.

**Ethical Concerns:**

["NO or VERY MINOR ethics concerns only"]

**Final Justification:**

I thank the reviewers for their careful response, which has cleared up many confusions. While I think the paper has substantial merit, I maintain my current score (which is leaning positive) because of the following standing limitations:
1. **Limited evaluation**: as pointed out by other reviewers as well, the paper only uses two Atari games to perform all of their evaluation. To convincingly show the paper's assumptions are not too narrow to be at least somewhat broadly applicable, I think a few more Atari games would go a long way.
2. **Clarity**: while I understand the reviewers cannot change the current pdf, I would argue the paper would greatly benefit from another pass on the writing. The biggest ticket here would be to incorporate the core parts of building the programmatic world model, which is currently in the appendix. Reviewer SJZQ has also pointed this out. In addition, things like including the original screenshots in Figure 3, clarifying how exactly the demonstrations were collected, etc. would all help push the paper to the next score.

**Limitations:**

yes

**Quality:**

3

**Strengths And Weaknesses:**

## Quality
I generally found the quality of the paper to be good. I verified all equations in the paper are correct, and I appreciate the experimental reporting includes error bands and other measures of uncertainty (though it would be great if the authors clarified what this is). In addition, I appreciate the authors going in and altering the environments to test for generalization.

Some things that could further improve the quality of the paper:
1. I understand getting this method to run even on a single Atari game requires substantial work. That being said, only two Atari games does feel on the low side. Maybe the authors could add a few environments from the Atari-5 subset: https://arxiv.org/abs/2210.02019.
2. In table 1, there are no baselines that also use the demonstration that is used for PoE. It would be good if the authors could include a behavioral cloning baseline as well as maybe one offline RL baseline that uses this demonstration.

## Clarity
I generally found the clarity of the paper to be lacking. One of the biggest concerns I have is that a lot of the core parts of building the programmatic world model is left to the appendix. I understand the authors would do this for the planning part which is not core to the contribution, but the way to construct the programs that are used as experts is completely missing in the main body of the paper. This feels strange as it is core to the method.

In addition, I found the paper lacking in clarity in the following (sometimes more minor) places:
1. Line 105: does "projection of $o$ onto feature $f$" mean "indexing $o$ with $f$"? From what I understood, $o$ is a vector of state attributes (e.g. velocity) and so $f$ just serves as an index into this vector? If this is the case, it might make sense to reword this to improve clarity.
2. I found the paragraph starting on line 106 a bit confusing. Could the authors make the argument being made here a bit more precise / formal? Do the authors mean instead of conditioning on $o_{1:t}$, using some latent variable $z_t$ to represent this? And then the issue would be that the dimensions of $z_t$ might not be interpretable anymore?
3. The hard constraints paragraph starting on line 114 doesn't feel expansive enough to understand what's really going on or why the hard constraints are necessary. Including an example of a hard constraint here would go a long way.
4. Are the multi-timestep predictions being discussed in equation 4 used anywhere? If not, I'd recommend leaving this part out and use the space it provides to address some of the other comments.
5. Line 144: "We then add noise to the distributions ...". What kind of noise?
6. Why does the equation on line 166 not use the same independence assumptions as the one on line 4?
7. Line 172: how exactly are the demonstrations created? Are they random agents? And if not, how much work went into creating these demonstrations? The details here could reflect on the practicality and assumptions of the method.
8. Figure 3: it would be helpful here if screenshots of the original environments were shown as well so it's easy to see the differences.
9. Are the numbers after +- in the various tables standard errors? Or standard deviations? Or something else? And how many seeds were used to compute these? And does each seed regenerate the programmatic experts?
10. In Table 2 and 3, how are the prediction accuracies defined? Are they exact match?
11. Table 4 reports "planning successes" but as far as I could tell this is never defined. What does "planning successes" mean?

## Significance
The results seem good, especially the ones on Montezuma's revenge are impressive! However, my biggest concern is that while the authors evaluate on Atari games, they end up working on a simplified state space, not on the direct pixel space. So part of my concern here is: how different is working on this "simplified" Atari setting from working on a complex grid world? Is it the dynamics? The number of objects? It would be good to get some sense here in order to understand how big of a step the authors are taking away from toy grid worlds.

## Originality
While there has been work on building world models with programs, it is certainly a challenge to get them to work on more complex environments. This combined with the modular expert program approach makes the paper feel pretty original. I haven't seen any other work trying to build world models in this way.

---

> ### Author Rebuttal · Authors · 2025-07-30
>
> Thank you for your positive feedback! We are glad the reviewer finds our Montezuma's Revenge result impressive. We address the raised concerns below:
>
> > Concern 1: My biggest concern is that while the authors evaluate on Atari games, they end up working on a simplified state space, not on the direct pixel space. So part of my concern here is: how different is working on this "simplified" Atari setting from working on a complex grid world? Is it the dynamics? The number of objects? It would be good to get some sense here in order to understand how big of a step the authors are taking away from toy grid worlds.
>
> **OCAtari is much “noisier” than gridworlds, and much higher-resolution, involving fine-grained motion and physics. It is also partially observed.** Handling partially observed stochastic environments with fine-grained physics is an important step toward making these techniques applicable in the real world.
>
> To illustrate the complexity of OCAtari, we include below two example trajectories from Pong. Here is a trajectory (displaying just the Pong paddle’s velocity y and the action) from a real Pong gameplay (Note that NOOP stands for NO OPeration):
>
> ```
> paddle velocity = 0
> -> action = RIGHT
> paddle velocity = -6
> -> action = RIGHT
> paddle velocity = -8
> -> action = NOOP
> paddle velocity = -15
> -> action = NOOP
> paddle velocity = -2
> -> action = NOOP
> paddle velocity = -2
> -> action = NOOP
> paddle velocity = 0
> ```
>
> And here is a trajectory in the same gameplay but at a different timestep:
>
>
> ```
> paddle velocity = 0
> -> action = RIGHT
> paddle velocity = -3
> -> action = RIGHT
> paddle velocity = -18
> -> action = NOOP
> paddle velocity = -8
> -> action = NOOP
> paddle velocity = -6
> -> action = NOOP
> paddle velocity = -1
> -> action = NOOP
> paddle velocity = 0
> ```
>
> Notice how the two trajectories lead to different sequences of velocities despite the same action sequences. In fact, in the first trajectory, performing NOOP when velocity = -2 can lead to two possibilities: velocity = 0 or -2. Performing NOOP also does not immediately stop the paddle from moving; the paddle only stops moving (velocity = 0) after a few timesteps.
>
> > Concern 2: One of the biggest concerns I have is that a lot of the core parts of building the programmatic world model is left to the appendix. I understand the authors would do this for the planning part which is not core to the contribution, but the way to construct the programs that are used as experts is completely missing in the main body of the paper. This feels strange as it is core to the method.
>
> Thank you for pointing this out. If accepted we will use our extra page to move those details to the main text.
>
> > Concern 3: Some things that could further improve the quality of the paper:
> > 1. I understand getting this method to run even on a single Atari game requires substantial work. That being said, only two Atari games does feel on the low side. Maybe the authors could add a few environments from the Atari-5 subset
> > 2. In table 1, there are no baselines that also use the demonstration that is used for PoE. It would be good if the authors could include a behavioral cloning baseline as well as maybe one offline RL baseline that uses this demonstration.
>
> We agree with the reviewer. We were not aware of Atari-5 when we wrote the paper, and it would have been a good addition to the paper. In table 1, WorldCoder also uses the same demonstration that is used for PoE-World, but yes, we use an on-policy algorithm (PPO) as a model-free RL baseline, so this baseline could not take advantage of the demonstration. An offline RL baseline may help with Pong, as the demonstration shows the player scoring once, but for Montezuma’s Revenge, the demonstration never shows positive score, so we do not expect off-policy RL to help in this case. We believe that the experimental results will convey the same message even with an off-policy RL baseline that uses the demonstration.
>
> > Concern 4: I found the paper lacking in clarity in the following (sometimes more minor) places:
> > 1. Line 105: does "projection of $o$ onto feature $f$" mean "indexing $o$ with $f$"? From what I understood, $o$ is a vector of state attributes (e.g. velocity) and so $f$ just serves as an index into this vector? If this is the case, it might make sense to reword this to improve clarity.
> > 2. I found the paragraph starting on line 106 a bit confusing. Could the authors make the argument being made here a bit more precise / formal? Do the authors mean instead of conditioning on $o_{1:t}$, using some latent variable $z_t$ to represent this?
> > 3. The hard constraints paragraph starting on line 114 doesn't feel expansive enough to understand what's really going on or why the hard constraints are necessary. Including an example of a hard constraint here would go a long way.
> > 4. Are the multi-timestep predictions being discussed in equation 4 used anywhere? If not, I'd recommend leaving this part out and use the space it provides to address some of the other comments.
> > 5. Line 144: "We then add noise to the distributions ...". What kind of noise?
> > 6. Why does the equation on line 166 not use the same independence assumptions as the one on line 4?
> > 7. Line 172: how exactly are the demonstrations created? Are they random agents? And if not, how much work went into creating these demonstrations? The details here could reflect on the practicality and assumptions of the method.
> > 8. Figure 3: it would be helpful here if screenshots of the original environments were shown as well so it's easy to see the differences.
> > 9. Are the numbers after +- in the various tables standard errors? Or standard deviations? Or something else? And how many seeds were used to compute these? And does each seed regenerate the programmatic experts?
> > 10. In Table 2 and 3, how are the prediction accuracies defined? Are they exact match?
> > 11. Table 4 reports "planning successes" but as far as I could tell this is never defined. What does "planning successes" mean?
>
> 1. Yes, we will reword it.
> 2. Yes, The POMDP formulation would condition on a latent variable $z_t$ instead of $o_{1:t}$. The issue is that $z_t$ would be an output of a learned component; it would be an output of $z_t = f_\phi (o_{1:t-1}, a_{1:t-1})$ where we are learning $f_\phi$. Since $f_\phi$ would be changing, $z_t$ would also change, and so all $p^{expert}_i$, which takes $z_t$ as input, needs to change correspondingly
>
> 3. We display the learned constraints in Table 10. An example is something like player_obj.bottom_side == conveyer_belt_obj.top_side
> 4. It is used in the PoE-World algorithm evaluated in the paper.
> 5. Small epsilon value: 1e-6.
> 6. Line 166 is meant to formalize how world models can be used in general (not specific to our work), so we did not include the independence assumption we use in equation 4 there.
> 7. They are not random agents. We do a few iterations of demonstrations to make sure it covers all the causal rules of the environments. The important thing here is that the demonstration shows examples of the causal rules needed to do well in the environment (for example, the player falls when its feet are not supported, touching the skull kills the player, etc.). Creating this demonstration is not hard for humans, but a random agent would not be sufficient.
> 8. Thank you, that is a good point. We will try to include the original screenshots in Figure 3 if there is enough space, if not, we will put them in the appendix.
> 9. They are standard errors. We use three seeds. Each seed does regenerate the programmatic experts :)
> 10. Yes, they are exact match
> 11. Thank you for pointing it out, we will fix it. Planning successes for Montezuma’s Revenge mean successfully grabbing the key.
>
> > Q1: In the definition of $Z_f$ in the equation after line 105, is it tractable to iterate over $o_f$? I guess the authors are using some discretized values with some range for velocity etc? And if not, how are continuous values handled?
>
> In OCAtari, the velocities are discrete, corresponding to per-pixel movement
>
> > Q2: On line 132, how expensive is it to repeat step 1 - 3 every time the observed trajectories get updated?
>
> Not expensive – we optimize the code so that it only synthesizes experts for newly added observations, and also we skip synthesizing experts for observations that are already well-explained by the current world model.
>
> > Q3: Line 140: "x/y velocity and visibility". Do the observations not include bounding boxes / positions of objects? And what does visibility mean?
>
> The world model takes as input an observation with bounding boxes and positions, and so it only needs to predict velocities and visibility (whether the object gets deleted). Object sizes do not change in Pong and Montezuma’s Revenge, so world models do not need to predict bounding boxes.
>
> > Q4: In Table 2 and 3, why are the prediction accuracies typically better on the alt environments? This feels a bit counterintuitive.
>
> The prediction accuracies are on random frames (generated by random actions). It is possible that random actions in the original environment may lead to unseen/unusual situations easier than in the alternative environments.
> > Q5: On line 226, are the 4000+ lines of code "active" or is most of it pruned? And also, how many experts does this typically consist of? It would be great if the authors could give some insight / statistics like this on the resulting world model programs.
>
> They are all active (the pruned programs are deleted from the world model and do not count words in the 4000+ line of code). There are usually around 300-500 experts.

---

> > ### Comment · Reviewer_LjAF · 2025-08-04
> > **Thank you**
> >
> > I thank the reviewers for their careful response, which has cleared up many confusions. While I think the paper has substantial merit, I maintain my current score because of the following standing limitations:
> > 1. **Limited evaluation**: as pointed out by other reviewers as well, the paper only uses two Atari games to perform all of their evaluation. To convincingly show the paper's assumptions are not too narrow to be at least somewhat broadly applicable, I think a few more Atari games would go a long way.
> > 2. **Clarity**: while I understand the reviewers cannot change the current pdf, I would argue the paper would greatly benefit from another pass on the writing. The biggest ticket here would be to incorporate the core parts of building the programmatic world model, which is currently in the appendix. Reviewer SJZQ has also pointed this out. In addition, things like including the original screenshots in Figure 3, clarifying how exactly the demonstrations were collected, etc. would all help push the paper to the next score.

---

> > > ### Author Response · Authors · 2025-08-04
> > >
> > > We thank the reviewer for engaging in this discussion, and we are glad that many confusions have been cleared.
> > >
> > > We acknowledge the two remaining concerns and would like to make quick notes about each of them to help the reviewer with the upcoming reviewer-AC discussion.
> > >
> > > - For limited evaluation, we would like to note that although we use two games in the paper, we also evaluate our method on the Alternate (Alt) version of each game, so in some sense, we have four environments in total.
> > >
> > > - For clarity, if our paper gets accepted, we would do more editing passes and use the extra page to move more technical details from the appendix to the main text for the camera-ready version of the paper.
> > >
> > > We thank the reviewer again for a positive review of our paper!

---

### Official Review · Reviewer_SJZQ · 2025-07-01

**Clarity:** 3
**Significance:** 3
**Originality:** 3
**Rating:** 4
**Confidence:** 3

**Summary:**

Learning a world model is crucial for building capable AI agents. This work focuses on how to leverage large language models to synthesize a world model utilizing weighted program experts. The program experts are synthesized utilizing a large language model. With only a few observations, this approach can capture the underlying rules of a complex, non-grid environment. The authors demonstrate their methodology on Atari’s Pong and Montezuma’s Revenge.

**Questions:**

1. How well does the learned world model reflect the actual dynamics of the environment?
2. How does the world model help downstream policy learning? Can you expand the scope of the Figure 5 results to show this more comprehensively?
3. Why does the method show different learning behaviors on Pong vs. Montezuma’s Revenge?
4. What does the discrepancy indicate for the general applicability and limitations of the approach?
5. In real-world settings, how would one identify or acquire the required objects, actions, and possible interaction results to build the program experts?
6. How are the expert programs synthesized?
7. How is the world domain defined and incorporated into the LLM?
8. What prompt strategy is used to guide the synthesis?

**Ethical Concerns:**

["NO or VERY MINOR ethics concerns only"]

**Final Justification:**

The authors has addressed all my concerns. I have raised my score accordingly.

**Limitations:**

Yes.

**Quality:**

2

**Strengths And Weaknesses:**

**Originality: 4/5**

This paper presents an innovative and refreshing approach to synthesizing an environment using a product of program experts. The generated rules effectively capture the semantics of the environment and enable learning from only a few observations.

**Quality: 2.5/5**

I have some concerns about the experimental setup. The main goals should be to demonstrate (1) how well the world model reflects the actual dynamics of the environment, and (2) how the world model supports the learning process of downstream models. However, I interpret Figure 4 primarily as evidence of data efficiency, without providing clear insights into research questions (1) or (2). On the other hand, Figure 5 presents interesting results relevant to research question (2), but it is limited in scope, as it shows only one learning policy in a single learning environment. Table 1 illustrates efficient adaptation to new environments, which is promising, but I believe Figure 5’s setup is a more direct response to research question (2). Since you have access to the ground truth setup, adding an explicit evaluation addressing research question (1) should be feasible. Additionally, extending the results shown in Figure 5 would strengthen the paper’s overall illustration of its core ideas.

Further, the Pong game and Montezuma’s Revenge exhibit very different behaviors in terms of learning: the PPO performs better in Pong, whereas in Montezuma’s Revenge the other methods fail to learn. While adding another experiment on a different game may not be feasible during the rebuttal period, I would appreciate more insight into why this discrepancy occurs and how it reflects on the general applicability and limitations of your approach.

**Significance: 3/5**

The world modeling is a crucial topic for AI agents building. While program synthesis could be an important building block. However, applying this work to general case could be challenging on how do people know the objects, actions, and potential interaction results? Can you elaborate further on how to get these building blocks in a real world scenario?

**Clarity: 3/5*

The paper reads well and the images are very helpful for illustration. However, can you elaborate more on how the expert programs are synthesized? Specifically, how is the world domain defined and passed into the large language model? What is the prompt strategy?

---

> ### Author Rebuttal · Authors · 2025-07-28
>
> Thank you for your feedback! We address the raised concerns below:
>
> > Concern 1: The main goals should be to demonstrate (1) how well the world model reflects the actual dynamics of the environment… However, I interpret Figure 4 primarily as evidence of data efficiency
>
> We believe **Table 2** (Next observation prediction accuracy vs baseline) and **Table 3** (Next observation’s object attributes prediction accuracy vs baseline) is what the reviewer is looking for.
>
> > Concern 2: (2) how the world model supports the learning process of downstream models… Figure 5 presents interesting results relevant to research question (2), but it is limited in scope
>
> **We consider a broader range of ways world models can support decision-making: Both learning (RL) AND planning.** Our main experiments, Table 1 and Figure 4, analyze planning, while Figure 5 shows how the world model supports downstream learning via RL.
>
> We focus more on planning than RL learning because planning is usually seen as having stricter requirements on the accuracy of world models, as errors can accumulate when an agent plans a long sequence of actions before acting. Pre-training model-free RL on world models (like Dreamer [1]) is usually easier to get to work and see gains of world models.
>
> > Concern 3: Further, the Pong game and Montezuma’s Revenge exhibit very different behaviors in terms of learning: the PPO performs better in Pong, whereas in Montezuma’s Revenge the other methods fail to learn.
>
> **This is because Montezuma’s Revenge is much harder than Pong. There is a consensus among RL researchers that Montezuma’s Revenge is one of the hardest Atari games for RL.** Go-Explore, Nature 2021 [2] calls it a *grand challenge*. Many papers explicitly mention Montezuma’s Revenge in their title/abstract [3, 4]. Reviewer LjAF notes that the Montezuma’s Revenge result is impressive.
>
> Model-free RL generally fails on Montezuma's Revenge, unless trained on an exorbitantly large number of environment interactions.
>
> > Concern 4: program synthesis could be an important building block. However, applying this work to a general case could be challenging on how do people know the objects, actions, and potential interaction results? Can you elaborate further on how to get these building blocks in a real world scenario?
>
> All these building blocks (perception, planning/acting, and world modeling) have their own dedicated research areas, and one paper cannot do everything at once. We believe that building AI agents that can learn, reason, and adapt flexibly in the real world requires progress on all of these building blocks, but in this work, we target the world modeling problem.
>
> Then, to get each of these building blocks to work in a real world scenario, we think researchers should gradually work on more and more complex domains over time. Previous work in symbolic world models only considered gridworlds and text worlds. We go beyond that and work on non-gridworlds like OCAtari: Easier than raw Atari, but much more complex than gridworlds. Ultimately, these games serve as testing grounds for research, and this work should be seen as a stepping stone toward more general agents.
>
> > Concern 5: Can you elaborate more on how the expert programs are synthesized? Specifically, how is the world domain defined and passed into the large language model? What is the prompt strategy?
>
> We have put details on how programs are synthesized and how input is turned into natural language to be passed into LLMs in Appendix A.1.
>
> > Q1: What does the discrepancy [between Pong and Montezuma’s Revenge] indicate for the general applicability and limitations of the approach?
>
> Montezuma’s Revenge is a long-horizon sparse-reward problem (see our reply to concern 3). The discrepancy indicates that model-free RL struggles in such settings (relative to planning). Although this is well known, it strengthens the argument for learning a world model: Only with such a model is planning possible.
>
> **References:**
>
> *[1] Hafner, D., Pasukonis, J., Ba, J., & Lillicrap, T. (2025). Mastering diverse control tasks through world models. Nature, 1-7.*
>
> *[2] Ecoffet, A., Huizinga, J., Lehman, J., Stanley, K. O., & Clune, J. (2021). First return, then explore. Nature, 590(7847), 580-586.*
>
> *[3] Salimans, T., & Chen, R. (2018). Learning Montezuma's revenge from a single demonstration. arXiv preprint arXiv:1812.03381.*
>
> *[4] Aytar, Y., Pfaff, T., Budden, D., Paine, T., Wang, Z., & De Freitas, N. (2018). Playing hard exploration games by watching youtube. Advances in neural information processing systems, 31.*

---

> > ### Comment · Reviewer_SJZQ · 2025-08-05
> >
> > > Concern 1: The main goals should be to demonstrate (1) how well the world model reflects the actual dynamics of the environment… However, I interpret Figure 4 primarily as evidence of data efficiency
> >
> > Besides table 2 and table 3, I wonder whether there is a more direct way comparing the ground truth rules against the actual rule in the model? For example, the ground truth has 300 rules, and among the 3000 rules that is generated by the LLM, what is the precision and the recall?
> >
> > > Concern 4: program synthesis could be an important building block. However, applying this work to a general case could be challenging on how do people know the objects, actions, and potential interaction results? Can you elaborate further on how to get these building blocks in a real world scenario?
> >
> > The major concern I have here is about the potential scaling problem. "World models produced by PoE-World consists of 4000+ lines of code for Montezuma’s Revenge compared to WorldCoder’s less than 100 lines. " Compare to the ground truth program, how much is the redundancy? Do you think scalability could be a problem, for example, extending this experiment to minecraft?
> >
> > I am happy to raise the score once the two concerns are further addressed. The other concerns are well addressed.

---

> ### Author Response · Authors · 2025-08-05
>
> We thank the reviewer for engaging in this discussion! We clarify the reviewer's questions below:
>
> > Besides table 2 and table 3, I wonder whether there is a more direct way comparing the ground truth rules against the actual rule in the model? For example, the ground truth has 300 rules, and among the 3000 rules that is generated by the LLM, what is the precision and the recall?
>
> There is a ground truth source code written in Assembly (as opposed to Python like our world model) for Atari 2600's Montezuma's Revenge (we are told not to paste links in rebuttal, but it is pretty easy to find this on Google search if the reviewer is curious). The program is 4721 lines long. **It is unclear, however, how to count how many "rules" there are, as the source code is written in a single Assembly program**. so we do not know if there is a good way of calculating rule precision and recall.
>
> In addition, **even if the ground truth source code could somehow be structured into rules, it is also possible that a single ground truth rule could be broken down and represented by multiple rules generated by LLMs**. For these reasons, it is hard to have an apples-to-apples comparison, and so we stick to next prediction accuracy and downstream learning metrics as an indicator of how good the learned world model is for this paper.
>
> > The major concern I have here is about the potential scaling problem. "World models produced by PoE-World consists of 4000+ lines of code for Montezuma’s Revenge compared to WorldCoder’s less than 100 lines. " Compare to the ground truth program, how much is the redundancy? Do you think scalability could be a problem, for example, extending this experiment to minecraft?
>
> **For world models produced by PoE-World, redundancy is handled by the following mechanism: once a behavior is well-explained by some experts, the algorithm would not synthesize more experts to explain that behavior.** Thus, there might be redundant experts if similar experts are all synthesized at the same timestep. However, in the next timesteps, the corresponding behavior would be well-explained, so our algorithm would not try to synthesize more experts to explain that behavior
>
> It is possible that one can try to manually code up a world model that contains less number of lines than PoE-World's world model, just like how a software engineer can refactor code to make it more compact. Directly removing experts in PoE-World's world models, nevertheless, will cause a drop in the likelihood value of observed trajectories. This is the result of how redundancy is handled as explained in the paragraph above -- PoE-World only adds experts if they would help increase likelihood of observed trajectories.
>
> In terms of scalability, **the runtime of the algorithm is linear with respect to the number of experts, so PoE-World should be able to scale well.** To build model-based agents for Minecraft, we suspect that other components of the agents, apart from world modeling, would present big obstacles: perception, decision-making, etc. We leave these other components for future work.
>
> Please do not hesitate to ask any follow-up questions!

---

> > ### Comment · Reviewer_SJZQ · 2025-08-05
> >
> > The authors has addressed all my concerns. I have raised my score accordingly.

---

> > > ### Author Response · Authors · 2025-08-06
> > >
> > > We thank the reviewer again for engaging in this discussion, and we are glad we are able to address all of the reviewer's concerns!

---

### Official Review · Reviewer_nRCp · 2025-07-01

**Clarity:** 3
**Significance:** 2
**Originality:** 3
**Rating:** 5
**Confidence:** 4

**Summary:**

This paper puts forth the idea of modeling the world (environment) as a product of (expert) programs.
Each program is some deterministic python code, and it predicts one attribute, or state variable, of
the environment. A collection of programs generates (zero or more) predictions for each state variable,
and these predictions can be combined to predict the most likely state of the environment. The weights
that are assigned to each Python program can be learned by finding which combination of weights generates
the predictions that best match the demonstrations.

Each Python program is generated using a LLM. The weights are learned by gradient descent. I am guessing
that some human effort goes into decomposing the world into its constituent state variables.

The paper shows that their approach of world modeling can yield useful models from just a few demonstrations,
so, there is a massive saving in the number of data points required. The environment here is taken from
two Atari games. The paper also shows benefits of having a product-of-symbolic-experts model for several
tasks, such as planning, or predicting the next observation, or RL.

**Questions:**

(1) Is the summary of the paper above accurate, or have I misrepresented the work?
(2) Are there cases where decomposing the world model as done in this paper is not a good idea? Even in classical control theory, when the world is modeled as, say, a linear system, dx/dt = Ax + Bu, where x is a vector of state variables, the world model is decomposed into constituent parts since each x_i is predicted separately using a different equation; however, the predictions are still conditioned on the current estimate, x, of the full state of the system. What is the input to the Python program -- I think it is a history of past observations, but I am not sure how long a history. I also understand that the input does not include the current estimate of the state of the system. Is that right?
(3) The same question as (1), but for the evaluation part of the review (titled the strengths and weaknesses).

**Ethical Concerns:**

["NO or VERY MINOR ethics concerns only"]

**Final Justification:**

No change to original score. My score is the most positive, perhaps it is a bit lenient. I found the approach of learning multiple small programs and combining them to get a world model interesting, and something that makes sense. The idea also has some validation from the past research going back many years. That is the main reason for keeping the positive rating. I do recognize the challenges in taking this work forward.

**Limitations:**

Yes.

**Paper Formatting Concerns:**

None.

**Quality:**

3

**Strengths And Weaknesses:**

The paper describes an interesting piece of work. The claims and results in the paper look reasonable.
The evaluation is only on a very small set of worlds, but the gains are clearly seen.
In some ways, this paper does a U-turn and goes back to the original way of designing systems in
systems theory. We first learn a plant (environment) model, and then use the plant model to design a
controller (planner). The first step is often referred to as system identification. Typically, one starts
with linear or nonlinear, discrete-time or continuous-time or hybrid, parametric models for the plant, and then
one learns the parameters from some demonstrations (system identification). Once we have a model of the
environment, one uses it to control or plan or do a host of other things. This paradigm has existed for a long
time and served us well. In contrast to this paradigm, a second approach has been to just learn a model that
performs the end-to-end task -- bypassing the step of learning a world model and directly learning to control.

The main innovation here is the idea of using multiple deterministic programs as the world model. There is also
some precedence to this idea. David Harel's work on scenario-based programming had a similar flavor. Of course,
it was before the time we had LLMs that could generate Python programs from observations.

The results on the "the role of hard constraints" are intriguing. I am not surprised that hard constraints
do not affect next observation prediction -- I can see that -- but it is surprising that they contribute to planning
success. Is the planner only using the learned model (learned using the hard constraints), or is it using
the learned model plus the hard constraints separately? In other words, are the hard constraints used in two
different places in the 2nd row of Table 2?

My overall impression is that the paper makes an interesting contribution that can be potentially useful
in some cases.

---

> ### Author Rebuttal · Authors · 2025-07-30
>
> Thank you for your positive feedback! We are glad the reviewer finds this work interesting. We answer the reviewer’s questions below:
>
> > Question 1: Are there cases where decomposing the world model as done in this paper is not a good idea? Even in classical control theory, when the world is modeled as, say, a linear system, dx/dt = Ax + Bu, where x is a vector of state variables, the world model is decomposed into constituent parts since each x_i is predicted separately using a different equation; however, the predictions are still conditioned on the current estimate, x, of the full state of the system. What is the input to the Python program -- I think it is a history of past observations, but I am not sure how long a history. I also understand that the input does not include the current estimate of the state of the system. Is that right?
>
> We think there are two different questions here. First, whether decomposing the world model is always a good idea; and second, whether conditioning predictions only on the past history (but not the other state variables) is too strong of an assumption.
>
> For the first question, we think decomposing the world model should never really hurt – having a single program as a world model is a special case  of PoE-World (with number of experts = 1).
>
> For the second question, this relates to a research direction we’re currently exploring. Factoring the state such that each state variable does not depend on the current values of other state variables works in this work, because we’re given the ground truth state variables (x/y position/velocity and visibility of each object).
>
> However, if we do not have access to these ground truth state variables and we have to *learn* the state variables, then state variables can become entangled (like “player y-velocity” correlates with “player’s feet supported by platform”), so the factored state assumption no longer holds. In our ongoing work, we’ve relaxed the independence assumption and added experts that model correlations between state variables. This gives interesting technical puzzles, as we cannot easily normalize the final probability distribution, so we need to use training techniques from the area of energy-based models (EBM) [1]; we’re currently exploring this direction in follow-up work.
>
>
> > Question 2: I am not surprised that hard constraints do not affect next observation prediction -- I can see that -- but it is surprising that they contribute to planning success. Is the planner only using the learned model (learned using the hard constraints), or is it using the learned model plus the hard constraints separately? In other words, are the hard constraints used in two different places in the 2nd row of Table 2?
>
> The planner does not use the hard constraints separately; it only uses the world models, whose outputs are affected by the hard constraints. We also find this somewhat surprising, and we suspect that hard constraints tend to play an important role when the world model has to generate a multi-timestep sequence of observations (as opposed to one-step prediction) where errors can accumulate.
>
> > Question 3: (1) Is the summary of the paper above accurate, or have I misrepresented the work?
>
> > And the same question as (1), but for the evaluation part of the review (titled the strengths and weaknesses).
>
> The written summary is a good and concise summary of our work :)
>
> The evaluation part is also accurate. We were not aware of the connection to older systems theory work, but the learning algorithm you described does sound very similar to what we have in this paper. We see this work as relating back to many older works but done in the modern age where LLMs can be used as powerful program synthesizers, and we demonstrate how such methods can work on a domain more complex than what symbolic world models have previously considered.
>
> **References:**
>
> *[1] Song, Y., & Kingma, D. P. (2021). How to train your energy-based models. arXiv preprint arXiv:2101.03288.*

---

> > ### Comment · Reviewer_nRCp · 2025-08-05
> >
> > Thank you for the answers. I have no further questions. The role of hard constraints is intriguing and perhaps worthy of some discussion in the revised version, if there is space.

---

> > > ### Author Response · Authors · 2025-08-06
> > >
> > > We thank the reviewer again for a positive and detailed review of our paper, and we are glad we are able to answer all of the reviewer's questions! As we promised with reviewer LjAF, we will do more editing passes and use the extra page to move more technical details in the appendix to the main text if this paper gets accepted.

---

### Official Review · Reviewer_RHFT · 2025-07-03

**Clarity:** 3
**Significance:** 3
**Originality:** 3
**Rating:** 5
**Confidence:** 4

**Summary:**

They introduce a world model that uses multiple LLMs to synthesize programs, then optimize on the weights of these programs to describe the rule set behind the observed trajectories. This allows to compose many small programs, as opposed to a large monolithic program to describe the domain, which allows for greater compositionality and extends to more complex domains.

**Questions:**

- An ask would be to push on compositionality in terms of more rigorous definitions as well as another domain more applicable to evaluating it.
- A detailed analysis in a domain with more complex underlying rules or is less object-centric and seeing what representation the programs will use in that case would allow me to build confidence about its ability to extend to more domains.
- Please expand on the partial observability point more. Especially having experiment results to back it up.

**Ethical Concerns:**

["NO or VERY MINOR ethics concerns only"]

**Final Justification:**

I had concerns about novelty and the difference between WorldCoder, but the rebuttal's addressed my concerns and have updated my view on the originality of this paper. Thus, I have updated my score to an Accept.

**Limitations:**

I would mention the reliance on object-centricity with one more line. "Symbolic programs expect symbolic inputs" is valid, but can be made more clear with a direct mention to where this approach does and doesn't apply.

**Quality:**

3

**Strengths And Weaknesses:**

Strengths:
- The algorithm is simple and clearly explained.
- There are clear applications and a need for an approach like this. In terms of program-structured world models, it seems to be the logical next step. I enjoy the references to human cognition.

Weaknesses:
- There is a lack of novelty of this approach. I see its promise but at its core its WorldCoder but with multiple smaller synthesizers that gets aggregated together.
- Even though compositionality is a major motivation behind the approach, there were no formal definitions for compositional generalization and experiments really emphasizing that. I wouldn't argue that the alterations made to the two environments is sufficient to show that. However, I do believe the algorithm could extend to more difficult, composition-specific domains.
- This approach would extend well to domains with large amounts of simple rules that are object-centric. However, when the rules cannot be broken down into such modular, small components, when the scene is not as distinct in terms of its labeled objects, or when the size of the rule set is less than m, I am unsure of how this approach will perform.
- There were mentions to partial observability but no justification or expansion on it. This is a key point that got completely overlooked

---

> ### Author Rebuttal · Authors · 2025-07-30
>
> Thank you for your feedback! We address the raised concerns below:
>
> > Concern 1: There is a lack of novelty of this approach. I see its promise but at its core its WorldCoder but with multiple smaller synthesizers that gets aggregated together.
>
> Quick clarification before the actual reply: it is the programs that get aggregated, not the synthesizers, but this is probably a typo from the reviewer. The actual reply is below:
>
> **We believe our novelty lies not in the idea of aggregation itself**, as we indeed cite many works advocating the idea of composing smaller parts to obtain a sophisticated whole. **Our novelty is in how the aggregation is done:** we turn small, deterministic programs into simple distributions and aggregate them stochastically to obtain sophisticated, stochastic world models. We highlight key differences between WorldCoder and PoE-World in a table below:
>
> | | WorldCoder | PoE-World (ours) |
> |---|---|---|
> | Learning mechanisms | Code generation (with refinement) with LLM | Code generation with LLM AND Gradient-based weight optimization |
> | Representation | Discrete | Mixed Discrete/Continuous |
> | Stochasticity | No, deterministic world models | Yes, fine-grained stochastic world models |
> | Scalability | Low, world models with ~100 lines of code | High, world models with ~4000 lines of code |
> | Most complex domains handled | Gridworlds, text domains | Non-gridworlds, i.e., OCAtari |
>
> > Concern 2: Even though compositionality is a major motivation behind the approach, there were no formal definitions for compositional generalization and experiments really emphasizing that.
>
> By “compositional generalization”, we mean two things:
> 1. Generalization to object combinations not seen during training: what Cosmos, ICLR’24 [1] calls “entity composition”
> 2. Factorization of the world model into rules conserved across object types: What Cosmos calls “relational composition”, and what RIMS [2] calls “independent mechanisms”.
>
> Experiments in Table 1 and Figure 4 evaluate both forms of compositionality: The Alt environments evaluate entity composition, and the comparison with the WorldCoder baseline evaluates relational composition. We will revise the paper to clarify this point about compositionality.
>
> > Concern 3: This approach would extend well to domains with large amounts of simple rules that are object-centric. However, when the rules cannot be broken down into such modular, small components, when the scene is not as distinct in terms of its labeled objects, or when the size of the rule set is less than m, I am unsure of how this approach will perform.
>
> **Although the experts in this work are relatively simple, they could be more complex: we can in fact even use WorldCoder wholesale to generate experts for PoE-World.** So even if the environment cannot be described by modular rules, our approach can just fall back to WorldCoder. We think though that many environments are mostly modular, and note that experts can model inter-object interactions.
>
> > Concern 4: There were mentions to partial observability but no justification or expansion on it. This is a key point that got completely overlooked
>
> **OCAtari, the domain we use in this work, is partially observed**: the current scene and the current action alone cannot describe the next scene. This is true *even for Pong.* Looking to the future, the real world is partially-observed, so handling this complexity is an important step for program world models. We will revise the paper to clarify that OCAtari is partially observed.
>
> **References:**
>
> *[1] Sehgal, A., Grayeli, A., Sun, J. J., & Chaudhuri, S. (2024). Neurosymbolic Grounding for Compositional World Models. ICLR.*
>
> *[2] Goyal, A., Lamb, A., Hoffmann, J., Sodhani, S., Levine, S., Bengio, Y., & Schölkopf, B. (2019). Recurrent independent mechanisms. arXiv preprint arXiv:1909.10893.*

---

### Official Review · Reviewer_aKQa · 2025-07-03

**Clarity:** 3
**Significance:** 2
**Originality:** 3
**Rating:** 4
**Confidence:** 4

**Summary:**

This paper introduces a method for learning world models (distributions over future observations given the past) as a product of experts, where each expert is a small Python program that attempts to capture a particular aspect of the world dynamics. The experts are generated by LLMs prompted with a small batch of trajectories. Then, a gradient-based optimizer fits weights to the experts in order to maximize the likelihood of the observed data. Experts with small weights are pruned, and the procedure can repeat while the agent collects more observations from the environment. Experiments on two Atari games, Montezuma Revenge and Atari, show that the learned world model, combined with a planning algorithm, performs significantly better than baselines under the small data regime. The world model is also shown to be useful to pre-train a PPO agent, improving its sample efficiency.

**Questions:**

* In (3) (L119), is the aggregation of hard constraints not supposed to be a conjunction (and) as opposed to disjunction (or)? If it is really a disjunction, why are they "hard" constraints if you only need one of them to be satisfied? (for instance, leaving the screen downwards wouldn't necessarily be a "hard" constraint unless all the constraints prevent it?)
* Did the authors observe any experts that become too slow to compute as the trajectory grows? Was the unpredictability of running times ever a problem for the scalability of the learned world model?
* I'm confused about the "Multi-timestep predictions" description. The text describes the idea here of assuming independence to phrase Multi-timestep predictions as a product of next-timestep experts. But Equation 4 contains $p^{expert}(o_{t+k} | o_{1:t}, a_{1:t})$. When k > 1, this is not a next-timestep expert anymore. Don't you need to also condition on an accumulating sequence of observations and actions (up to $t+k-1$)?
* L157: serve "as" a simulator?
* How often do the synthesized experts overlap in behavior? Is there any pressure for them to be orthogonal? What happens if the same (useful) expert is generated multiple times - do all of its copies get kept by the weight-based pruning procedure, since they'd have similar weights?
* Are there other existing environments with less challenging perception where PoE-World would be able to scale better than WorldCoder in terms of the complexity of the world model? I think those could be compelling cases to study, even if the observations are not at the pixel level, and would help address concerns about the generality of the method.
* How large does the world model get on Pong? Intuitively, this looks like an environment where the dynamics are simple enough for a monolithic program to capture (even if it might not generalize as well to Pong-Alt). Is the main advantage here against WorldCoder the fact that the learning method is compositional, even if the total size of the world model might not be as large as in MR?

**Ethical Concerns:**

["NO or VERY MINOR ethics concerns only"]

**Final Justification:**

The authors provided a clear response to most of my concerns. I agree that the world modelling results are interesting and potentially inspiring for other domains, even though their application in particular to Atari problems creates complications that are orthogonal to world modelling per se (as the authors discussed), and thus make the results a bit harder to disentangle than I think is ideal. That said, these are challenging domains and the method is sound, so I recommend acceptance.

**Limitations:**

Yes

**Quality:**

2

**Strengths And Weaknesses:**

The paper proposes a novel method that significantly improves the scalability of learning programmatic world models. The scale of the learned world models (4k+ lines of code) is encouraging for this line of work. The idea of using deterministic expert programs, but treating their output as probability distributions and using gradient-based optimization to learn how to weight the experts, and prune them, is technically interesting. The experiments are well-motivated, and the choices of baselines make sense.

My main high-level concerns are about how specific to Montezuma Revenge (and Pong) were some of the choices that the authors made.

First, the paper assumes an existing object detector. This seems quite hard to obtain for most rich domains, and certainly will be a major challenge if we want to eventually extend this line of work to naturalistic settings. The authors note that even for Atari games having an accurate scene parser is difficult. Thus, while I understand that the paper focuses on world modeling and not on perception, the dependency on an object detector might limit how broadly applicable the method ends up being.

It also seems like the planner might be hard to generalize across domains. In Step 1 in A.2, it's unclear how you obtain the list of all possible abstract states. Is it assumed that all of the relevant objects are given a priori, and that they do not change (e.g., objects appear or disappear)? I'd have liked to see a discussion of which of these assumptions might be violated in other cases (e.g., even in other Atari games), and which are supposed to be reasonable in a wider range of environments.

Overall, I like the idea and the set of experiments, but I'm unclear on how applicable the method currently is in other environments (Atari and otherwise) - how hard it would be to satisfy its assumptions, and whether it could bring gains in cases where perception is not necessarily a significant challenge (but world modeling can still be, regardless). This can be improved by expanding the evaluation in terms of new environments

---

> ### Author Rebuttal · Authors · 2025-07-28
>
> Thank you for your feedback! We address the raised concerns below:
>
> > Concern 1: The reviewer is concerned that we assume an object detector and a planner, limiting generalization to other domains. The reviewer wonders how hard it is to satisfy these assumptions. (See concern 1.1 and 1.2 for specific assumptions)
>
> We agree with the reviewer’s concerns on the limitations of used object detectors and planners. It is indeed hard to obtain these assumed components. **We note, however, that these assumptions are not specific to our work and should be seen as generic limitations of symbolic world models line of research.** We elaborate on each assumption below:
>
> > Concern 1.1: The paper assumes an existing object detector
>
> Object detectors are an important assumption, and one often made in robot decision-making. All our baselines use the same structured input representation, rather than pixels. Under this comparison, we make significant progress relative to previous symbolic world models (Worldcoder NeurIPS 24 [1], Code World Models NeurIPS 24 [2]). **While WorldCoder and CWM, both published at NeurIPS 24, work on gridworlds and text domains, we work on OCAtari (Atari + Object detector), which is much more complex.**
>
> Other works learn symbolic world models from pixels [3], but do not scale to dynamics as complex as those we consider. We hope that publishing our paper paves the way for combining our method with such works, which would help address this concern.
>
> > Concern 1.2: seems like the planner might be hard to generalize across domains
>
> Planning is a very hard problem with its own research community. **The planner used is not part of our method/contribution: we use the same planner for both our work and the baseline (WorldCoder) we are comparing against. In the paper, we denote our agent as PoE-World + Planner to emphasize this point.** We further note that a planner is not always needed: As figure 5 shows, we can run model-free RL (PPO) in the world model and see performance gains. We acknowledge that we use a more domain-specific planner, but this was necessary to check that our world models were accurate enough for long-range planning.
>
>
>
> > Concern 2: Whether it (this work) could bring gains in cases where perception is not necessarily a significant challenge (but world modeling can still be, regardless)
>
> Thanks for pointing this out: We focus on world modeling because it still presents a significant challenge even when perception and planning are taken as given, as evidenced by the poor performance of the baselines. Our main experiments (Table 1 and Figure 4) serve to demonstrate that **given reliable perception, PoE-World outperforms other methods, even in the presence of fine-grained physics, stochasticity, and partial observability.**
>
> > Q1: Are there other existing environments with less challenging perception where PoE-World would be able to scale better than WorldCoder in terms of the complexity of the world model?
>
> Yes! Robot decision-making, specifically Task and Motion Planning (TAMP), is a prime example. We think Atari is adequate for a first paper, though: Montezuma’s Revenge has served as the centerpiece of several well-known papers [4, 5].
>
> > Q2: How large does the world model get on Pong? ... Is the main advantage here against WorldCoder the fact that the learning method is compositional, even if the total size of the world model might not be as large as in MR?
>
> Pong has surprisingly complex physics, requiring a world model with ~3000 lines of code. Compositional factoring allows learning independent programs for each object, and even breaking the dynamics of a single object up into several programs.
>
> To illustrate the complexity of Pong, we include below two example trajectories from Pong. Here is a trajectory (displaying just the paddle’s velocity y and the action) from a real Pong gameplay (Note that NOOP stands for NO OPeration):
>
> ```
> paddle velocity = 0
> -> action = RIGHT
> paddle velocity = -6
> -> action = RIGHT
> paddle velocity = -8
> -> action = NOOP
> paddle velocity = -15
> -> action = NOOP
> paddle velocity = -2
> -> action = NOOP
> paddle velocity = -2
> -> action = NOOP
> paddle velocity = 0
> ```
>
> And here is a trajectory in the same gameplay but at a different timestep:
>
>
> ```
> paddle velocity = 0
> -> action = RIGHT
> paddle velocity = -3
> -> action = RIGHT
> paddle velocity = -18
> -> action = NOOP
> paddle velocity = -8
> -> action = NOOP
> paddle velocity = -6
> -> action = NOOP
> paddle velocity = -1
> -> action = NOOP
> paddle velocity = 0
> ```
>
> Notice how the two trajectories lead to different sequences of velocities despite the same action sequences. In fact, in the first trajectory, performing NOOP when velocity = -2 can lead to two possibilities: velocity = 0 or -2. Performing NOOP also does not immediately stop the paddle from moving; the paddle only stops moving (velocity = 0) after a few timesteps.
>
> > Q3: How often do the synthesized experts overlap in behavior? Is there any pressure for them to be orthogonal? What happens if the same (useful) expert is generated multiple times - do all of its copies get kept by the weight-based pruning procedure, since they'd have similar weights?
>
> All copies do get kept, so there might be redundant experts if similar experts are all synthesized at the same timestep. However, in the next timesteps, the corresponding behavior would be well-explained, so our algorithm would not try to synthesize more experts to explain that behavior.
>
> > Q4: I'm confused about the "Multi-timestep predictions" description. The text describes the idea here of assuming independence to phrase Multi-timestep predictions as a product of next-timestep experts. But Equation 4 contains $p^{expert}(o_{t+k}|o_{1:t}, a_{1:t})$. When $k > 1$, this is not a next-timestep expert anymore. Don't you need to also condition on an accumulating sequence of observations and actions (up to $t+k-1$)?
>
> Yes, for multi-timestep experts, we make an extra assumption, so we do not condition the predictions on the sequence of observations and actions between time $t$ and $t + k$.
>
> > Q5: Did the authors observe any experts that become too slow to compute as the trajectory grows? Was the unpredictability of running times ever a problem for the scalability of the learned world model?
>
> Runtime is linear with the number of experts, so it is quite predictable. The experts tend to be quite simple, so they run fast. Appendix A5 includes runtime.
>
> > Q6: In (3) (L119), is the aggregation of hard constraints not supposed to be a conjunction (and) as opposed to disjunction (or)? If it is really a disjunction, why are they "hard" constraints if you only need one of them to be satisfied? (for instance, leaving the screen downwards wouldn't necessarily be a "hard" constraint unless all the constraints prevent it?)
>
> As discussed in appendix A.1, in Equation (3), we choose to use a disjunction rather than a conjunction because the physics in video games can be peculiar and unrealistic—a player might have their body overlap with a platform when climbing down a ladder attached to that platform. The constraints are still hard in the sense that each of them is deterministic, not probabilistic. In certain situations, we agree that a conjunction could also be a reasonable choice.
>
> **References:**
>
> *[1] Tang, H., Key, D., & Ellis, K. (2024). Worldcoder, a model-based llm agent: Building world models by writing code and interacting with the environment. Advances in Neural Information Processing Systems, 37, 70148-70212.*
>
> *[2] Dainese, N., Merler, M., Alakuijala, M., & Marttinen, P. (2024). Generating code world models with large language models guided by monte carlo tree search. Advances in Neural Information Processing Systems, 37, 60429-60474.*
>
> *[3] Liang, Y., Kumar, N., Tang, H., Weller, A., Tenenbaum, J. B., Silver, T., ... & Ellis, K. (2024). Visualpredicator: Learning abstract world models with neuro-symbolic predicates for robot planning. arXiv preprint arXiv:2410.23156.*
>
> *[4] Kulkarni, T. D., Narasimhan, K., Saeedi, A., & Tenenbaum, J. (2016). Hierarchical deep reinforcement learning: Integrating temporal abstraction and intrinsic motivation. Advances in neural information processing systems, 29.*
>
> *[5] Ecoffet, A., Huizinga, J., Lehman, J., Stanley, K. O., & Clune, J. (2021). First return, then explore. Nature, 590(7847), 580-586.*

---

> > ### Comment · Reviewer_aKQa · 2025-08-04
> >
> > Thank you for the direct response to each of my points.
> >
> > Overall, I can see that the results are solid and in two domains where world modelling is a significant challenge. I get the point that the additional challenges related to perception, although required to test the method on OCAtari, are orthogonal to the paper's contributions. I do think that it would still be interesting to find domains that particularly isolate the challenges of world modelling, and that don't require the additional perception-specific components, might provide a clearer picture of the advantages of the PoE approach for world modelling. Some ideas for future work besides TAMP could also be the recent environments for Web/Mobile agents, where I expect there to be interesting world modelling challenges (e.g., modelling what buttons do to the visible / invisible application state), with long-range planning, and without the need for complex perception, or stochasticity. That said, I agree that the Atari domains are already interesting for a first paper exploring this approach. I thus revised my score.
> >
> > > Pong has surprisingly complex physics, requiring a world model with ~3000 lines of code.
> >
> > This is indeed surprising. Does "required" just mean that this is the size of the expert that PoE-World synthesized, or do the authors think that if you were to hand-write a similar world model as a monolithic program it would have a similar size? (it would be helpful if there was a "ground truth" implementation of the world model alone to compare to, but I assume there's none for Atari games).
> >
> > > Runtime is linear with the number of experts, so it is quite predictable. The experts tend to be quite simple, so they run fast.
> >
> > I just meant that the experts per se could have high asymptotic complexity (e.g., O(n^4) with n objects in obj_list), as they are general Python programs. But it sounds like this was not an issue as the experts tended to be simple in practice.

---

> ### Author Response · Authors · 2025-08-05
>
> We thank the reviewer for engaging in this discussion, and we are glad that we have helped clarify some of the reviewer's concerns!
>
> > This is indeed surprising. Does "required" just mean that this is the size of the expert that PoE-World synthesized, or do the authors think that if you were to hand-write a similar world model as a monolithic program it would have a similar size? (it would be helpful if there was a "ground truth" implementation of the world model alone to compare to, but I assume there's none for Atari games).
>
> There is in fact a ground truth implementation for Atari 2600's Montezuma's Revenge that we found online (we could not find it for Pong). It is written in Assembly, and the total line count is 4721 lines (we are told not to paste links in the rebuttal, but it is pretty easy to find this on Google if the reviewer is curious).
>
> We do suspect that one can manually hand-write a similar world model with a smaller size, just like how a software engineer can refactor code to make it more compact. At the same time, we do note that for PoE-World's world models, directly removing an expert from a world model would cause a drop in the likelihood value of the observed trajectories. At each point, PoE-World only adds experts if it helps increase the likelihood value of the observed trajectories (as explained in our reply to Q3)

---

> > ### Comment · Reviewer_aKQa · 2025-08-06
> >
> > Thanks for the pointer. I believe I did find the Assembly code with the same number of lines. However, I was more interested in an implementation of just a world model for comparison (i.e., with the same type signature as what PoE-world learns - though I don't think that would already exist for these games): it seems that the 4721 lines of Assembly also include everything from raw data (sprites, etc), to rendering logic, sound, and all other components of the full game. It looks like only around 1500 lines are for game logic, and being in Assembly, I'd assume that an equivalent Python implementation would likely be shorter. But again, this is perhaps not a useful point of comparison since this is implementing the dynamics, not a world model.
> >
> > My main question with that would be (as other reviewers pointed out) to understand the "rate of progress" with which the data gets explained. It could perhaps be possible to keep increasing the likelihood by adding hyper-specialized experts that only marginally help the likelihood but that don't capture the general rules in the world? (e.g., that condition on specific values of irrelevant variables for a given rule). In any case, I see this more as a consideration for future work to consider in order to improve scalability towards much more complex worlds (e.g. Minecraft, as pointed out in another discussion), in case there's anything interesting to note there.

---

> > > ### Author Response · Authors · 2025-08-06
> > >
> > > Thank you so much for the reply -- this is really helpful.
> > >
> > > We agree with everything the reviewer stated here. First, we agree that it is generally hard to do an apples-to-apples comparison with a ground truth world model. And the task of coming up with a world model that fits observations well is quite different from the task of coding up a new game / environment. Second, we also agree that it would be helpful to understand the rate of progress with which the data gets explained as we tackle even more challenging domains. While we only need around ~300 experts for OCAtari, we might need much more than that for more complex domains like robotics domains, and so understanding the rate of progress will become crucial. We will certainly keep this in mind for our future work, thank you!

---

### Decision · Program_Chairs · 2025-09-17

**Decision:**

Accept (spotlight)

**Comment:**

This paper focuses on an interesting direction - program-based world models. Specifically, the paper proposes to construct many small programs which are composable and uses an expert model with learnable weights to predict the future. The paper proposes a clearly novel paradigm and distinguishes itself from other standard world models. The symbolic nature can make it more data efficient. There are potential concerns of the paper such as bespoking and generalizability (Reivewer aKQa), experiments setups (reviewer SJZQ) and its evaluation on the symbolic world (reiviewer LjAF). With the more capable LLMs, exploring a programmable world model can have some nice impacts, hence I recommended acceptance.